# Actin polymerisation and crosslinking drive left-right asymmetry in single cell and cell collectives

Yee Han Tee [1] ✉, Wei Jia Goh[1,7], Xianbin Yong [1,7], Hui Ting Ong[1], Jinrong Hu[1], Ignacius Yan Yun Tay[1], Shidong Shi[1], Salma Jalal[1], Samuel F. H. Barnett[1], Pakorn Kanchanawong [1,2], Wenmao Huang[3], Jie Yan [1,3], Yong Ann Ben Lim[1], Visalatchi Thiagarajan[1], Alex Mogilner [4,5] & Alexander D. Bershadsky [1,6] ✉

Deviations from mirror symmetry in the development of bilateral organisms are common but the mechanisms of initial symmetry breaking are insufficiently understood. The actin cytoskeleton of individual cells self-organises in a chiral manner, but the molecular players involved remain essentially unidentified and the relationship between chirality of an individual cell and cell collectives is unclear. Here, we analysed self-organisation of the chiral actin cytoskeleton in individual cells on circular or elliptical patterns, and collective cell alignment in confined microcultures. Screening based on deep-learning analysis of actin patterns identified actin polymerisation regulators, depletion of which suppresses chirality (mDia1) or reverses chirality direction (profilin1 and CapZβ). The reversed chirality is mDia1-independent but requires the function of actin-crosslinker α−actinin1. A robust correlation between the effects of a variety of actin assembly regulators on chirality of individual cells and cell collectives is revealed. Thus, actin-driven cell chirality may underlie tissue and organ asymmetry.

While many animals demonstrate approximate bilateral symmetry, many important features of the body layout such as position of visceral organs, as well as the shape of organs themselves are usually asymmetric. These asymmetries are tightly programmed and aberrations in such a program can lead to severe defects in embryonic development[1]. Several examples of left-right asymmetry emerging have been discovered. At the level of entire organisms, emergence of left-right asymmetry has been thoroughly described for visceral organs in vertebrates[1,2] and formation of asymmetric body in snails[3]. The processes of asymmetric organogenesis include heart-looping in vertebrates[4], chiral shaping of hindgut and genitalia[5,6], and asymmetric tilting of wing bristles in *Drosophila*[7]. Cell groups confined to micropatterned adhesive substrates (in the form of stripes or rings) demonstrated chiral cell alignment and movement[8–10]. Finally, on the single cell level, the processes of intracellular swirling, cortical flow, and cell migration can demonstrate left-right asymmetry[11–15].

It is commonly believed that mechanisms underlying emergence of left-right asymmetry in diverse biological systems are based on the function of special chiral molecules[16], and in particular the chiral cytoskeletal fibres[17]. Indeed, several classes of cytoskeletal proteins were shown to be involved in the processes of left-right asymmetry development listed above. While the asymmetric positioning of visceral organs depends on numerous cilia-related proteins[18] and attributed to cilia function in specialised cells located in the embryonic node

[1]Mechanobiology Institute, National University of Singapore, Singapore 117411, Singapore. [2]Department of Biomedical Engineering, National University of Singapore, Singapore 117583, Singapore. [3]Department of Physics, National University of Singapore, Singapore 117542, Singapore. [4]Courant Institute, New York University, New York, NY 10012, USA. [5]Department of Biology, New York University, New York, NY 10012, USA. [6]Department of Molecular Cell Biology, Weizmann Institute of Science, Rehovot 7610001, Israel. [7]These authors contributed equally: Wei Jia Goh, Xianbin Yong. ✉e-mail: teeyeehan@nus.edu.sg; alexander.bershadsky@weizmann.ac.il

(left-right organiser), actin cytoskeleton-related proteins are involved in other examples of asymmetry. In particular, non-conventional myosins 1d and 1c are needed for asymmetric hindgut and male genitalia development in *Drosophila*[5,6], and myosin 1d is sufficient to induce chirality in other *Drosophila* organs[19]. We previously proposed the role of formin family proteins in the development of actin cytoskeleton chirality[11] and several recent publications established the role of diaphanous formin in dextral snail chirality[20,21], Daam formin in *Drosophila* hindgut and genitalia chirality[22], and *Caenorhabditis elegans* CYK-1 formin in chiral cortical flow of *C.elegans* zygote[23]. Finally, some data indicated the involvement of the actin filament crosslinking protein α−actinin1 in emergence of chirality in cultured individual cells and multi-cell spheroids[11,24]. However, how intrinsic chirality of the actin filaments[25] is translated in vivo into chirality of cells, and whether this is sufficient to explain the emerging of chirality in multicellular groups such as tissue and organs remains obscure.

In particular, it is unknown whether left-right asymmetry emerges as a result of the activity of a single chiral determinant such as myosin 1d[19] (perhaps different proteins in different systems) or is mediated by coordinated activities of a group of proteins with complementary functions. Here, we addressed this question by systematic investigation of the involvement of major actin-associated proteins in the regulation of left-right asymmetry of the actin cytoskeleton in individual cells. In the course of this analysis, we examined the effects of depletion of chirality regulators found in previous studies, namely formins, myosin 1c and 1d, and α−actinin1. We show that, even in this simple system, the chiral swirling of actin depends on the functions of several groups of actin regulators. We reveal several types of such regulators: depletion of some of them reduced or abolished chirality, while depletion of others reversed the direction of chirality.

The discovery of numerous regulators of chiral morphogenesis in individual cells allowed us to perform detailed comparison between factors affecting individual and collective cell chirality. It was previously unclear whether chiral asymmetry of the actin cytoskeleton in individual cells is related to emergence of collective chirality in cell groups. Here, we found that the majority of treatments affecting the asymmetric self-organisation of the actin cytoskeleton in individual cells also affected the asymmetric alignment of cell groups. In particular, all factors that reversed direction of cytoskeletal chirality in individual cells also reversed the direction of chiral cell alignment in cell groups. Altogether our findings provide the background for the future understanding of the processes of emerging left-right asymmetry in tissues and organs.

## Results

### Assessment of chiral organisation of radial fibres
We have previously shown that human fibroblasts (HFF) plated on fibronectin-coated circular islands with an area of 1800 μm$^2$ formed a chiral pattern of organisation of actin filament bundles. Radial actin fibres originating from focal adhesions at the cell edge eventually tilted to the right from the axis connecting the focal adhesion with the cell centre. This coincided with the development of chiral swirling − an anti-clockwise bias in centripetal movement of transverse fibres along the tilted radial actin fibres[11] (Supplementary Movie 1). To quantitatively investigate the molecular requirements for actin cytoskeleton chiral self-organisation, we introduced a quantitative method of assessment of radial fibre tilting (Fig. 1a to h, and Supplementary Fig. 1). First, we segmented the radial fibres using a deep-learning network (Unet-ResNet50) (Fig. 1a, b, e and f, and Supplementary Fig. 1a). The identified radial fibres are presented as having similar brightness (grey values) irrespective to their fluorescence intensity in the original image (Supplementary Fig. 1b). The segmented images were further binarized and skeletonised. Second, the image of circular cell was subdivided into eight concentric annular rings located at given distances from the cell edge (Fig. 1c and g, and Supplementary Fig. 1c). The mean tilt of

radial fibre segments located in each ring was calculated for each cell (Fig. 1d and h, and Supplementary Fig. 1c). The radial fibre pattern for the cell population was then characterised by the curve showing the mean tilt of radial fibre segments (averaged over all cells) as a function of their distance from the cell edge (Fig. 1l). In addition, to further characterise the variability between cells, we compared histograms characterising the distribution of the average radial fibre tilt in the single annulus located between 6−10 or 8−12 microns from the cell edge (Fig. 1i to k). Details of the methodology are described in methods section.

### Chiral pattern formation requires mDia1, Arp2/3 and cofilins 1&2
We evaluated the effects of the knockdowns of 10 of 15 formin family members expressed in fibroblasts on chiral organisation of radial fibres (Supplementary Fig. 2a to f). Knockdown of diaphanous-related formin, mDia1, did not prevent the formation of either radial or transverse fibres, but abolished the tilting of the radial fibres (Fig. 1j). As a result, mDia1 depleted cells exhibited a radially symmetric organisation of the actin cytoskeleton (Fig. 1j and l). The inhibitory effect of mDia1 knockdown on chiral actin pattern formation can be rescued by expression of exogenous mDia1-GFP construct (Fig. 1k and l, and Supplementary Fig. 2d and Supplementary Table 1, lines 1–3). The statistical analysis of the results of knockdown of several other abundant formins revealed that their effect on chirality can be classified into two groups (Supplementary Fig. 2b and c, and Supplementary Table 1, lines 124–133). FMNL2, FHOD3 and Daam1 knockdowns reduced the degree of actin cytoskeleton chirality, albeit not to such extent as knockdown of mDia1 (Supplementary Fig. 2b and Supplementary Table 1, lines 124–127 and 134–136). The knockdowns of other examined formins (FMN2, mDia2, mDia3, INF2, FHOD1 and Daam2) did not have an apparent effect on chirality (Supplementary Fig. 2c and Supplementary Table 1, lines 128–133).

We further studied the cell orientation in multicellular microcultures on rectangular adhesive islands with a 1:2 aspect ratio (300 × 600 μm). At 48 h following plating, the cells approached confluency and aligned mainly along the diagonal of the rectangles as seen from the orientation of nuclei or local average cell orientation characterised by nematic directors on phase-contrast images[26] (Fig. 2a to c and Supplementary Movie 2). Thus, each rectangular microculture was characterised by the mean angle between local nematic directors and the long axis of the rectangle (Fig. 2b and d). In complementary set of measurements, the long axes of elliptical nuclei were used instead of nematic directors (Fig. 2c and d). In control microcultures, the cells orientation was chiral so that the distribution of the values of angles characterising individual rectangles was asymmetric (Fig. 2d). In other words, the cells on the rectangular pattern preferentially aligned into a И-orientation rather than a N-orientation (when observed from above). We also examined rectangles with other aspect ratios and found that the 300 × 600 μm size was optimal for asymmetric alignment of microcultures (Supplementary Fig. 3).

We found that knockdown of mDia1 converted the distribution of the angles characterising individual microcultures into bimodal distribution (Fig. 2d and Supplementary Fig. 4), meaning that mDia1 knockdown cells in the rectangular microcultures co-oriented in И or N fashion with equal probability. Examining the knockdowns of other formins revealed that FMNL2, FHOD3, Daam1 knockdown significantly reduced the mean angles of nematic directors (Supplementary Fig. 2g and Supplementary Table 1, lines 148–150), while INF2 knockdown increased it (Supplementary Fig. 2h and Supplementary Table 1, line 154). Knockdowns of other formins – FMN2, mDia3, mDia2, FHOD1 and Daam2, did not produce significant effect (Supplementary Fig. 2h and Supplementary Table 1, lines 151–153 and 155–156).

We further assessed the effect of the knockdowns of other major regulators of actin polymerisation[27]. The suppression of the actin-

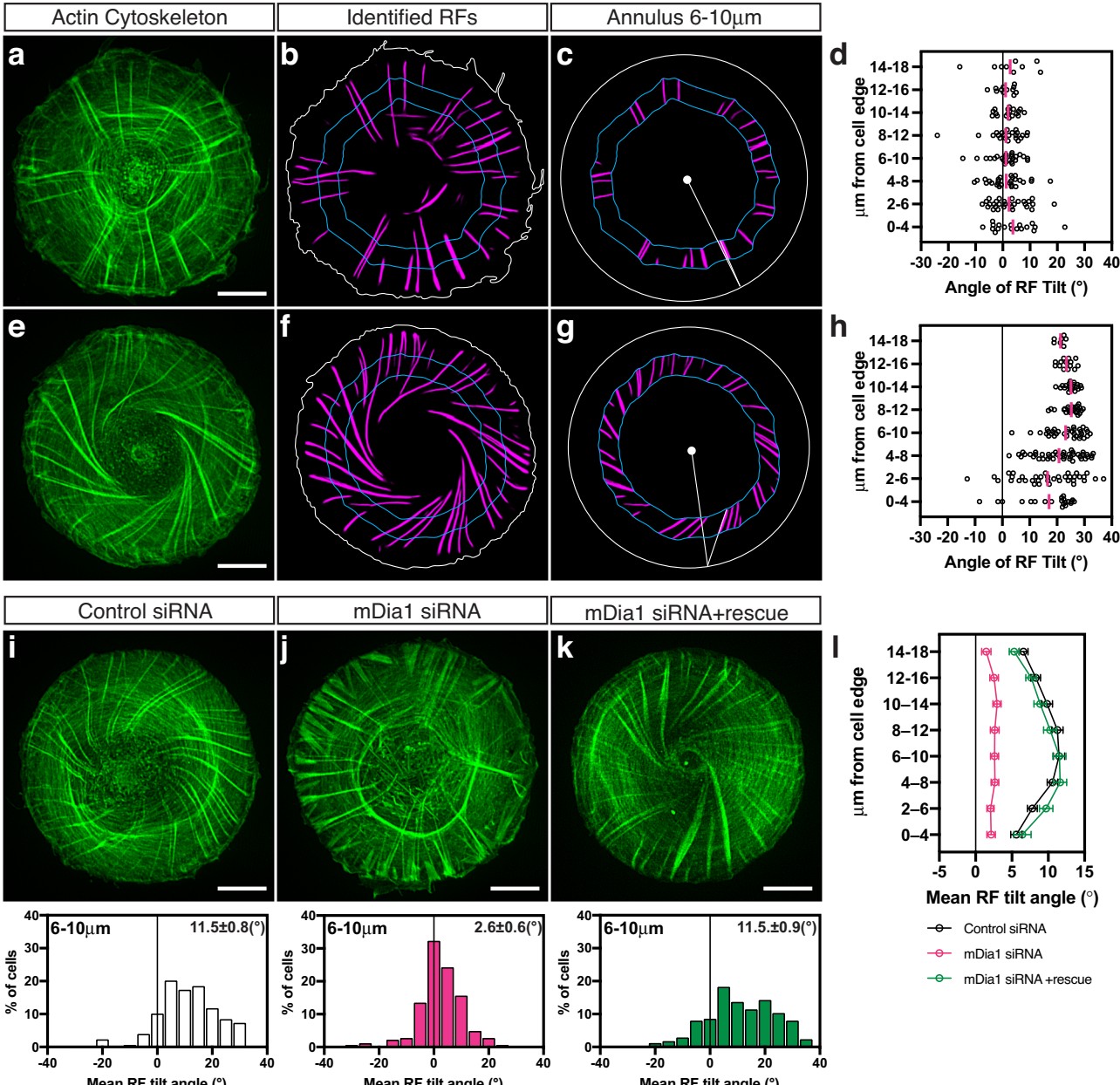

**Fig. 1 | Quantification of radial fibre tilt reveals a decrease in actin cytoskeleton chirality in mDia1 knockdown cells. a–h** Illustration of measurements of the radial fibre (RF) tilt. Selected fluorescence images of actin cytoskeleton in a radial (**a**) and a chiral (**e**) cells from the population of control cells fixed and stained with phalloidin at 6 h after plating. **b, f** RFs were identified by deep-learning procedure. Pair of cyan lines located at the 6 and 10 μm distances from the cell edge (white line) delineate the concentric belt that we termed the 6–10 μm annulus. Other annuli are defined in a similar way. The tilts of all RF segments in the annuli located at given distance from the cell edge were measured as shown in (**c**) and (**g**). In these measurements, the cell edge was replaced with best fit circle (white line) with the centre at the cell centroid (white dot). See Methods. See also Supplementary Fig. 1c. **d, h** The tilt values in all eight annuli of cells (**a**) and (**e**) are shown in (**d**) and (**h**) respectively. Each symbol represents the tilt angle of an individual RF segment. The mean of tilt values at each given distance from the cell edge are shown in magenta.

**i–l** Effect of mDia1 knockdown and rescue on RF tilt. **i–k** Typical examples of actin organisation visualised by phalloidin-staining of cells transfected with control siRNA (**i**), mDia1 siRNA (**j**) and mDia1 siRNA plus mDia1 full-length plasmid (**k**) 6 h following cell plating on circular patterns. The histograms in (**i–k**) show the distribution of mean RF tilt in the 6–10 μm annulus in cells under corresponding conditions (white bars – control siRNA-treated cells, magenta bars – mDia1 siRNA-treated cells, green bars – mDia1 siRNA-treated cells rescued by mDia1 plasmid). Histograms and mean±SEM values (in **i–k**) are based on measurements of 179 control cells, 186 mDia1 knockdown cells, and 177 mDia1 knockdown cells rescued by mDia1 plasmid overexpression. **l** Graphs of average RFs tilts (mean±SEM) as a function of the distance of annuli from the cell edge (the same cell samples as **i–k**). Scale bars, 10 μm (**a**, **e**, **i–k**). For statistical analysis, see Supplementary Table 1, lines 1–6.

nucleating Arp2/3 complex via the knockdown of its major component ARPC2 (Supplementary Fig. 5f), resulted in the inhibition of actin cytoskeleton chirality as evident by the reduction in radial fibre tilt angle (Supplementary Fig. 5a and b, and Supplementary Table 1, line 167). Otherwise, the overall effect of ARPC2 downregulation on the

actin cytoskeleton appearance in these experiments was relatively mild and cell spreading was not impaired.

The double knockdown of actin depolymerising proteins, cofilins 1 and 2 (Supplementary Fig. 5g), led to a pronounced inhibition of actin chirality (Supplementary Fig. 5c and d, and Supplementary Table 1, line

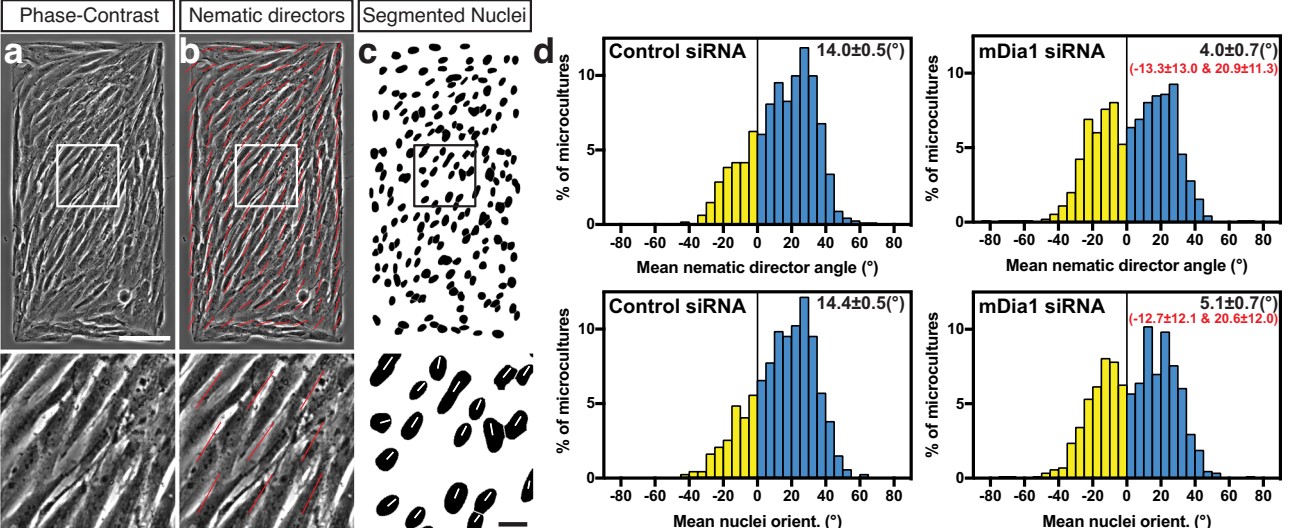

**Fig. 2 | Quantification of chiral alignment of cells in confined cell groups reveals a decrease of chirality in mDia1 knockdown cells. a** Phase-contrast image of cells 48 h following plating on rectangular adhesive pattern (300 × 600 μm). **b** Image shown in (**a**) overlaid with red lines representing average local orientation of cells (nematic directors). **c** Segmented Hoechst 33342-stained nuclei of cells shown in (**a**). Boxed areas in (**a** to **c**) are shown at higher magnification in the lower row. **d** Histograms showing distributions of the values of mean nematic directors angles (upper row) and mean nuclei orientation angles (lower row) characterising individual microcultures at 48 h after plating. The histograms were built based on average nematic director angles from 1530 control and 894 mDia1 knockdown microcultures, and average nuclei orientation angles from 1521 control and 844 mDia1 knockdown microcultures. Mean ± SEM values are indicated at the top right corner of each histogram. Negative and positive values are coloured in yellow and cyan respectively. Note that for mDia1 knockdown cells both nematic directors and nuclei orientation distribution are bimodal as determined by fitting it as a sum of 2 Gaussian distributions (see Supplementary Fig. 4); the values of the respective two means(±SD) are indicated in red. Scale bars, 100 μm (upper row); 20 μm (lower row). For statistical analysis, see Supplementary Table 1, lines 7–14.

169). The knockdown of related protein, actin depolymerising factor (ADF) (Supplementary Fig. 5g), diminished anti-clockwise actin cytoskeleton chirality slightly (Supplementary Fig. 5d and e, and Supplementary Table 1, line 172), but combined knockdown of ADF and cofilins 1 and 2 completely abolished cell chirality resulting in symmetrical distribution of radial fibre tilt with a zero mean value (Supplementary Fig. 5d and e, and Supplementary Table 1, lines 173 and 177). Re-introduction of only cofilin-1 to the cofilins 1 and 2 depleted cells was sufficient to restore anticlockwise chirality of the cell (Supplementary Fig. 5d, e and g, and Supplementary Table 1, line 171).

Investigation of the effects of depletion of ARP2/3 and cofilins/ADF family members on collective cell alignment in rectangular microculture revealed that, unlike the knockdown of mDia1, the knockdown of ARPC2, cofilins and ADFs only slightly affected the degree of chiral orientation of cells in microcultures on rectangles (Supplementary Fig. 5h and Supplementary Table 1, lines 179–182). It can be explained by assumption that knockdowns of ARPC2 and cofilins delayed the anti-clockwise chirality onset but not suppressed it entirely.

Lastly, the knockdowns of Ena/VASP family activators of actin filament elongation, VASP and Mena, moderately albeit significantly reduced the development of asymmetric actin pattern in individual cells (Supplementary Fig. 6a to e, and Supplementary Table 1, lines 188–189) but did not perturb the chiral alignment of cell collectives in microcultures (Supplementary Fig. 6f and Supplementary Table 1, lines 192–193).

**Perturbing actin polymerisation can alter chirality direction**
While depletion of mDia1, ARPC2, or cofilins reduced the asymmetry of actin pattern, the manipulations of several other actin-associated proteins or actin pharmacological perturbations reversed the direction of chirality (Fig. 3). Profilins are abundant actin-monomer sequestering proteins which can either augment or inhibit actin polymerisation, depending on their biological context[28]. We found that the siRNA-mediated knockdown of profilin 1 (Supplementary Fig. 7a to c)

reversed the direction of actin swirling in cells, resulting in sinistral chirality pattern (Fig. 3b and f, and Supplementary Movie 3). This effect can be rescued by the expression of exogenous human profilin 1 (Fig. 3c and f, and Supplementary Fig. 7b and Supplementary Table 1, line 17). The curve showing the average tilt of radial fibres as a function of distance from the cell edge in profilin 1 siRNA-treated cells was approximately a mirror-image of the curve for control cells (Fig. 3f). In our experiments, siRNA transfection only partially inhibited the expression of profilin 1 protein (0.38 ± 0.17 fold change) (Fig. 5h and Supplementary Fig. 7b and c) which, however, was sufficient to reverse chirality direction. At the same time, depletion of ~90% of endogenous profilin 2 (Supplementary Fig. 7b) not only did not reverse chirality direction but increased the degree of anti-clockwise chirality in terms of radial fibre tilt (Fig. 3d and f and Supplementary Table 1, line 18).

Capping proteins bind to the barbed end of actin filaments and interferes with both polymerisation and depolymerisation[27]. Knockdown of CapZβ, a subunit of capping protein CapZ, (Supplementary Fig. 7d) resulted in the reversal of direction of radial fibres tilting so that sinistral actin pattern was formed (Fig. 3e and f).

Of note, even though the effects of knockdowns of CapZβ, mDia1 and cofilins 1&2 were reproducible in experiments where actin pattern was quantified in fixed phalloidin-stained cells, they were less pronounced in experiments where actin was visualised by expression of LifeAct fused with either GFP or mRuby fluorescent proteins. The reason for these discrepancies is unknown; it might be related to some effects of LifeAct on actin polymerisation[29].

By screening of actin polymerisation affecting drugs, we found that latrunculin A, which sequester G-actin monomer and depolymerise F-actin[30], when applied at low concentration (20 nM), effectively reversed the direction of chirality – inducing sinistral actin pattern (Fig. 3g). The time course of chirality development in the presence of latrunculin A was the same as in control, while the average tilt of the radial fibres was of similar magnitude but opposite in direction as compared to control (Fig. 3h). The effect of latrunculin A was readily seen in both phalloidin-stained fixed cells and LifeAct-labelled live cells.

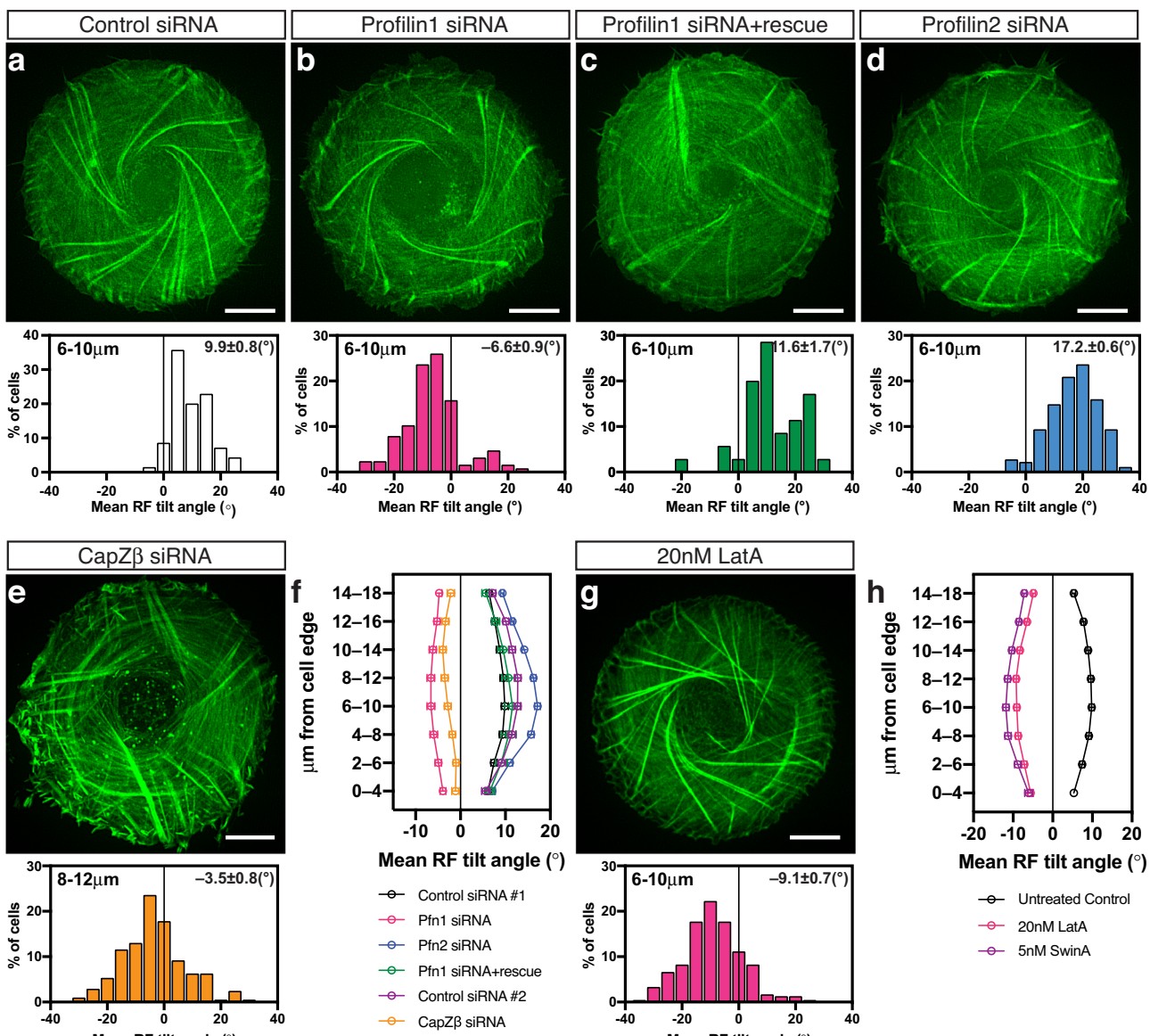

**Fig. 3 | Genetic knockdowns and pharmacological treatments reversing the direction of the actin cytoskeleton chirality. a–d** Typical examples of actin organisation visualised by LifeAct-GFP in control siRNA (**a**), profilin 1 (Pfn1) siRNA (**b**), profilin 1 siRNA plus Pfn1 full-length plasmid (**c**) and profilin 2 (Pfn2) siRNA (**d**) transfected cells. The histograms in (**a–d**) show the distribution of mean RF tilt in the 6–10 μm annulus in cells treated as indicated and imaged for 12–16 h after plating. Histograms and mean±SEM values are based on measurements during the entire period of imaging of 70 control cells, 127 Pfn1 knockdown cells, 35 Pfn1 knockdown cells rescued by co-transfection with full-length Pfn1 plasmid and 182 Pfn2 knockdown cells . **e** Typical example of actin organisation visualised by phalloidin-staining in CapZβ siRNA-transfected cells fixed at 6 h after cell plating. Histogram under the image shows the distribution of mean RF tilt in the 8–12 μm annulus in cells at 6 h after plating. Histogram and mean±SEM value are based on 208 cells. **f** Average values of RF tilts (mean±SEM) as a function of the distance of annuli from the cell edge for profilins and CapZβ experiments (the same cell samples as **a–e**). Control siRNA #1 and #2 represent the 70 and 146 control cells used in experiments with profilins and CapZβ respectively. **g** Typical example of actin organisation visualised by phalloidin-labelling in cells treated with 20 nM of latrunculin A (LatA). Histogram under the image shows the distribution of mean RF tilt in the 6–10 μm annulus of cells fixed at 6 h after cell plating. Histogram and mean ±SEM value are based on 243 cells. **h** Average values of RF tilts (mean±SEM) as a function of the distance of annuli from the cell edge for untreated control cells (*n* = 274), 20 nM LatA-treated cells (*n* = 243), and 5 nM swinholide A (SwinA)-treated cells (*n* = 153) at 6 h after cell plating. Colour coding in histograms (**b–e**) and (**g**) correspond to those indicated in graphs (**f**) and (**h**) respectively. Scale bars, 10 μm (**a–e** and **g**). See also Supplementary Movie 3. For statistical analysis, see Supplementary Table 1, lines 15–29.

Addition of latrunculin A reversed the direction of swirling in cells with an established anti-clockwise (dextral) actin cytoskeleton (Supplementary Movie 4). These effects were rapid and became evident within 1-2 h following latrunculin A addition. Washing out of latrunculin A resulted in rapid return to anti-clockwise actin swirling (Supplementary Movie 5). Among other actin polymerisation affecting drugs, the actin filament severing and dimer forming drug swinholide A[31], at 5 nM, produced the same reversing effect on chirality direction as latrunculin A (Fig. 3h). The fast effects of latrunculin A on chirality direction suggest that its

action does not depend on any transcriptional effects. Indeed, we showed that chiral tilting of radial fibres and anti-clockwise actin swirling, as well as reversion of direction of these processes upon latrunculin A treatment can occur in enucleated cells (Supplementary Fig. 8 and Movie 6). The experiment with enucleated cells also showed that tilting of the radial fibres does not dependent on their possible interaction with the nucleus.

The genetic knockdowns and drug treatments that changed the direction of chirality of the actin cytoskeleton in individual cells

induced corresponding changes in the chirality of alignment in microcultures on a rectangular pattern. Depletion of profilin 1 or CapZβ, or treatment with latrunculin A or swinholide A resulted in preferential cell alignment in the direction mirror-symmetrical to that of control cells (Fig. 4a and b). In all these cases, the average orientation of multicellular groups in rectangles was tilted at negative angles relative to the long axis of the rectangle (Fig. 4c and Supplementary Fig. 7e). Similar to results in individual cells, the knockdown of profilin 2 did not change the direction of cell alignment as compared to control (Fig. 4c and Supplementary Fig. 7e). Therefore, the effect of treatments affecting the direction of chirality in individual cells strongly correlates with their effect on the chirality of cell alignment in microcultures.

Altogether, the results presented in this section showed that human fibroblasts can demonstrate two types of actin cytoskeleton chirality – anti-clockwise, observed in wildtype cells, and clockwise, observed upon knockdowns of some actin regulating proteins and pharmacological treatments. Such situation is typical for some living systems, for example, pond snails demonstrate dextral (prevalent) and sinistral (rare) coiling. Since formin mDia1 is shown to be required for dextral anti-clockwise chirality (Figs. 1i to l and 5d and g), we examined its possible involvement in establishing sinistral clockwise chirality induced by profilin 1 knockdown and latrunculin A treatment. The data presented in Fig. 5 show that while knockdown of mDia1 suppressed the dextral chirality of fibroblast, it does not significantly affect the development of sinistral chirality (Supplementary Table 1, lines 48–49).

## Alteration of chirality direction is α−actinin1-dependent

Overexpression of actin crosslinking protein α−actinin1 resulted in the reduction of cell chirality and reversal of the direction of actin cytoskeleton swirling in a fraction of transfected cells in agreement with our previous study[11] (Fig. 6a and b, and Supplementary Fig. 9a). The effect of chirality reversal upon overexpression of α−actinin1 was also evident on multicellular microcultures confined to rectangular pattern (Fig. 6c and d, and Supplementary Fig. 9e). Overexpression of α−actinin4 largely reduced the fraction of chiral cells, while overexpression of another actin crosslinker, Filamin A, had no effect (Fig. 6b and Supplementary Fig. 9, b and c and Supplementary Table 1, lines 57, 58 and 61). Neither α−actinin1 knockdown alone nor the expression of truncated α-actinin construct, ABDdel-actinin[32], which interfered with the actin crosslinking activity of α−actinin, changed the chirality direction in individual cells and cell groups (Fig. 6e to h, Fig. 7a and e, and Supplementary Fig. 9d and e).

In view of the involvement of α−actinin1 in reversal of cell chirality in these and other study[24], we further investigated the combined effect of α−actinin1 loss-of-function and experimental manipulations, which reversed chirality direction. The average tilt of RFs in cells with double knockdowns of profilin 1 and α−actinin1 does not statistically significantly differ from that of control (Fig. 7b and Supplementary Table 1, lines 83–85). Double knockdowns of CapZβ and α−actinin1 resulted in dextral anti-clockwise chirality (Fig. 7c and Supplementary Table 1, lines 89–91). Latrunculin A treatment of α−actinin1 knockdown cells or ABDdel-actinin expressing cells (Fig. 7d and Supplementary Fig. 9f) resulted in broad symmetrical distribution of radial fibre tilt with the mean close to zero (Supplementary Table 1, lines 97 and 218). At least for α−actinin1 knockdown cells treated with latrunculin A, this distribution can be approximated by a sum of 2 Gaussian distributions (Supplementary Fig. 4b). Thus, reversal of chirality by depletion of either profilin 1 or CapZβ, or latrunculin treatment requires α−actinin1 function. In microcultures, knockdown of α−actinin1 similarly prevents the reversal of chirality upon profilin 1 (Fig. 7f and Supplementary Table 1, lines 101–103) or CapZβ (Fig. 7g and Supplementary Table 1, lines 104–106) depletion, but not upon latrunculin A treatment (Fig. 7h and Supplementary Table 1, lines 107–109).

Since several studies have clearly demonstrated the importance of myosin 1c and 1d in the chiral morphogenesis of *Drosophila* organs[5,6,19], we checked whether human myosin 1c and 1d are involved in individual and collective chirality development in our experimental systems. The differences between control and myosin 1c or myosin 1d knockdown cells (Supplementary Fig. 10d and e) in the development of the chiral actin pattern in individual cells were not statistically significant (Supplementary Fig. 10a to c and Supplementary Table 1, lines 219–220). However, chirality of collective alignment in microcultures of myosin 1c knockdown cells was significantly more pronounced than in control microcultures (Supplementary Fig. 10f and Supplementary Table 1, line 223). Thus, myosin 1c may participate in chirality regulation in our experimental system while such function of myosin 1d was not revealed. Of note, the human myosins 1c and 1d are not the exact homologs of their *Drosophila* counterparts with similar names[33], however, human myosin 1c did demonstrate chiral interaction with the actin filaments in an in vitro assay[34].

In order to assess whether any of the knockdown effects observed above could be attributed to an associated alteration in the expression level of other proteins involved in chirality regulation, we examined the transcriptional profile of cells depleted of major proteins strongly associated with chirality phenotype (Supplementary Fig. 11a). We found that, with few exceptions, knockdowns of the members of this group of proteins (mDia1, ARPC2, cofilins 1&2, CapZβ, profilin 1, and α−actinin1) only slightly, if at all, affected the expressions of other members of the group. Moreover, some increase in α−actinin1 transcriptional level upon knockdowns of mDia1 or cofilins 1&2 (Supplementary Fig. 11a) was not accompanied by an increase in its protein level (Supplementary Fig. 11b and c). These data suggested that phenotypic changes observed upon knockdown of these proteins are not mediated by transcriptional regulation of the expression of other members of this group. This is consistent with the apparent transcription-independent effect on chirality observed in latrunculin A-treated enucleated cells, as mentioned above (Supplementary Fig. 8).

## Correlation between individual and collective cell chirality

Altogether, in our experiments, we characterised the effects of 35 different pharmacological and genetic manipulations on the emergence of left-right actin cytoskeleton asymmetry in individual confined cells, as well as on chiral cell alignment of confined multicellular groups. To analyse the interrelationship between the establishment of left-right asymmetry in these two systems, we plotted the average angle between the nematic directors characterising the alignment of cells in cell groups versus the average tilt of radial fibre segments located between 6–10 microns from the cell edge (Fig. 8 and Supplementary Table 2). In spite of some discrepancies, as mentioned earlier, the correlation between these two parameters was highly significant (Pearson correlation coefficient, $r = 0.8104$, $p < 0.0001$) (Fig. 8). Ranking the data in ascending order also revealed a strong correlation between these two parameters (Spearman's rank correlation coefficient, $r = 0.7081$, $p < 0.0001$) (Supplementary Fig. 12a and Supplementary Table 2). Moreover, such presentation of the data clearly shows the correlation between effects of knockdown of majority of formin family members on individual and collective cell chirality (Supplementary Fig. 12b). Altogether these data demonstrate the role of actin cytoskeleton asymmetry in individual cells in the establishment of collective asymmetry of cell alignment in cell groups.

## Chirality of actin cytoskeleton in confined elliptical cells

In view of the obvious correlation between the chirality of actin cytoskeleton swirling in individual cells and the chiral nematic orientation of elongated cells in cell groups, we decided to check whether actin cytoskeleton in elongated cells could demonstrate chiral

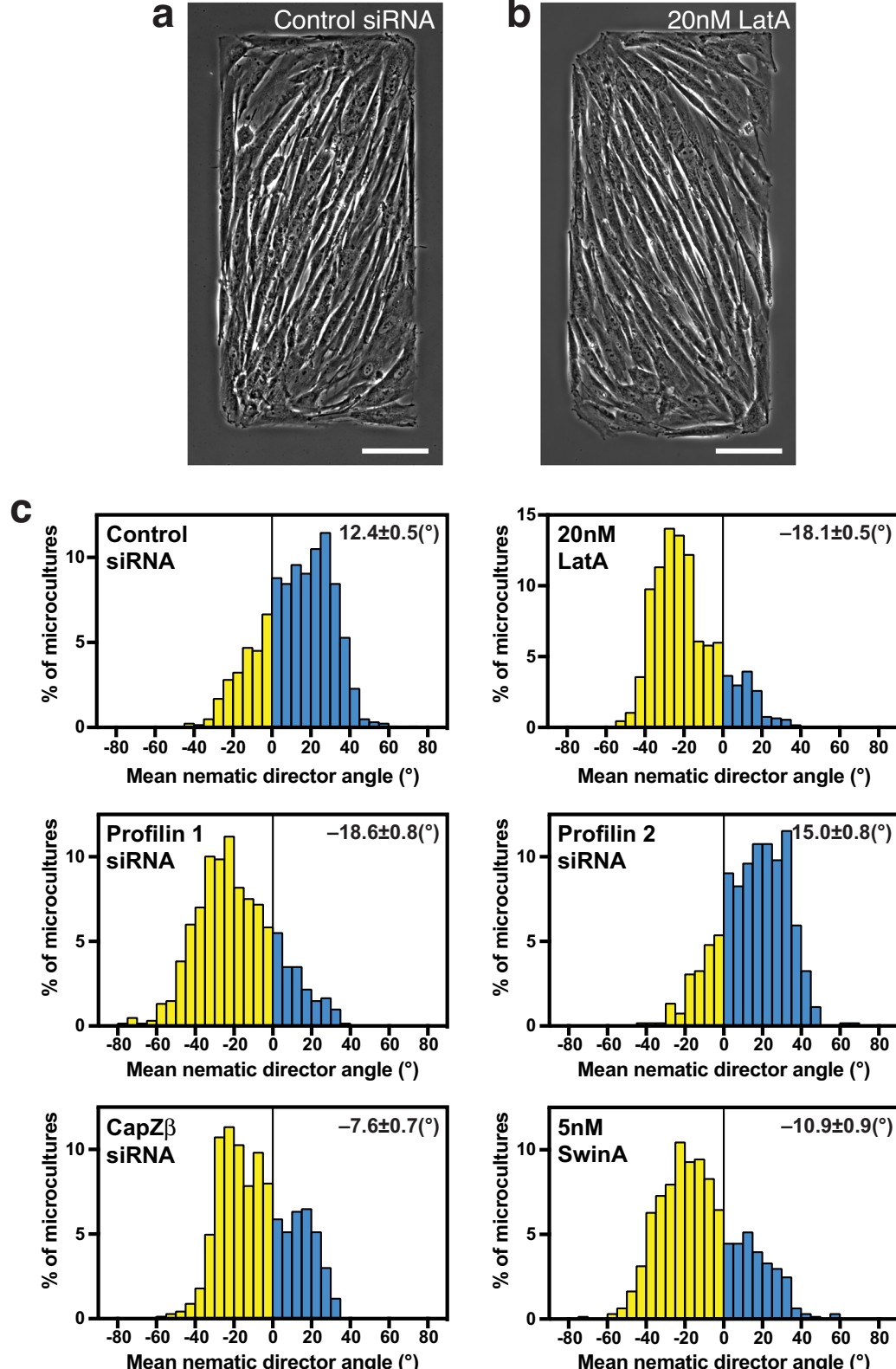

**Fig. 4 | Knockdowns of profilin 1 and CapZβ and treatments with latrunculin A and swinholide A reverse the sign of cell orientation angle in microcultures confined to rectangular micropatterns. a, b** Typical phase-contrast images showing dextral alignment of control cells (**a**) and sinistral (reversed) alignment of latrunculin A (LatA)-treated cells (**b**) 48 h following plating on rectangular adhesive pattern. **c** Histograms show distributions of the values of mean nematic directors characterising individual microcultures on rectangles for cells treated as indicated. Negative and positive values are coloured in yellow and cyan respectively. Mean ±SEM values are indicated at the top right corner of each histogram. The histograms were built based on average local cell orientation (nematic directors) values from 1168 control, 1031 LatA-treated, 597 Pfn1 knockdown, 519 Pfn2 knockdown, 661 CapZβ knockdown and 602 swinholide A (SwinA)-treated microcultures. See also Supplementary Fig. 7. Scale bars, 100 μm (**a, b**). For statistical analysis, see Supplementary Table 1, lines 30–40.

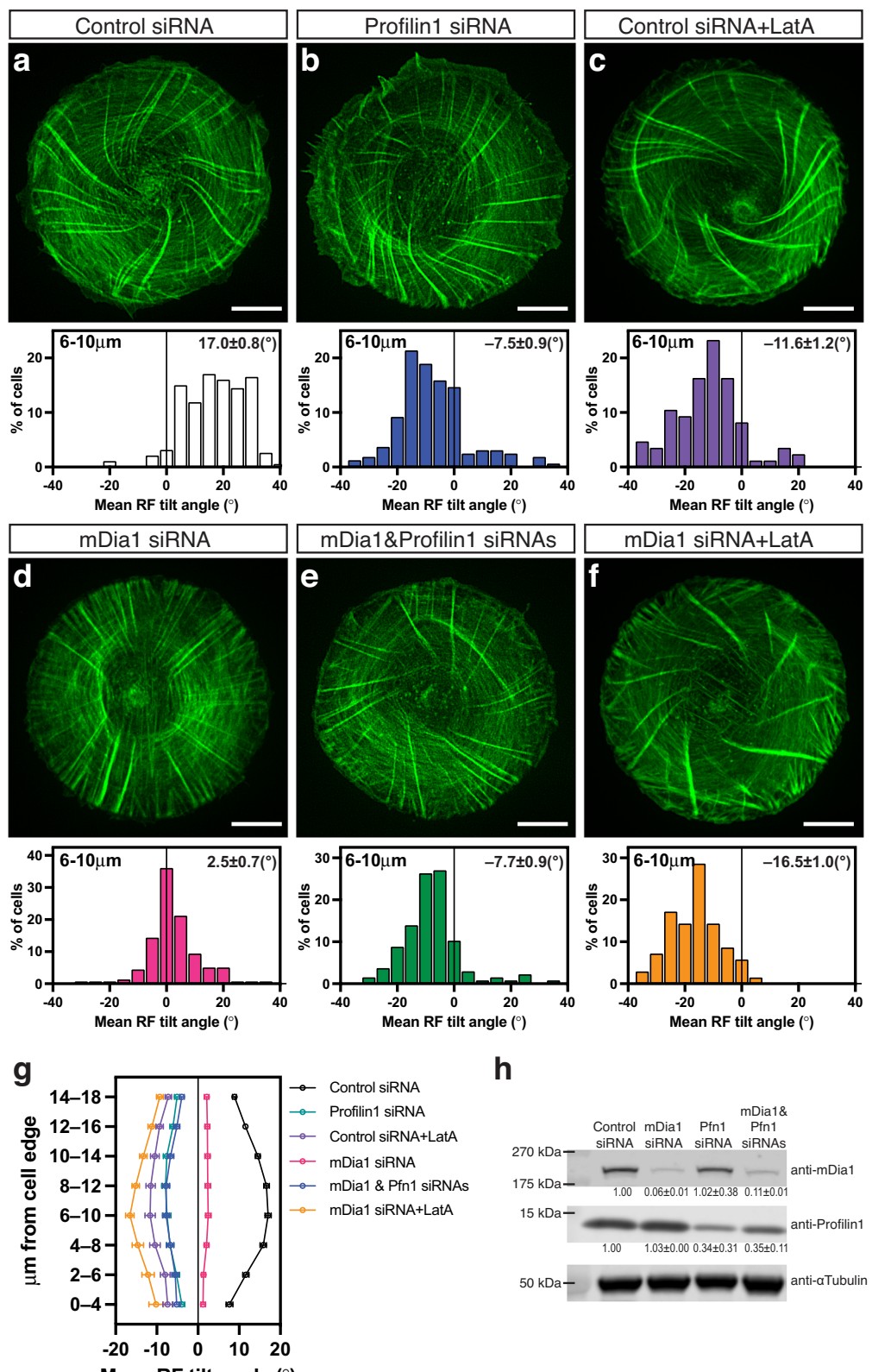

organisation. To this end, we plated individual cells on elliptical micropatterns with different aspect ratios (Fig. 9 and Supplementary Fig. 13). On circular patterns, the actin cytoskeleton evolves from the system of radial fibres through the chiral tilting of radial fibres and anti-clockwise swirling of actin flow to the linear organisation of parallel stress fibres (actin bundles containing myosin filaments and associated with focal adhesions at both ends) filling the entire cell[11]. During spreading on elliptical micropatterns, the growth of radial fibres and focal adhesions were stronger at the vertex regions characterised by higher curvature than at the sides of the ellipses (Supplementary Fig. 13 and Movie 7). The dynamic observations of cells revealed a chiral pattern of radial fibres and anti-clockwise swirling, similar to that in a circular cell but geometrically transformed to accommodate the elliptical shape (Fig. 9g and Supplementary Movie 7). Sometimes, even

**Fig. 5 | Formin mDia1 is dispensable for the development of clockwise (sinistral) actin cytoskeleton chirality in profilin 1 knockdown and latrunculin A-treated cells. a–f** Typical examples of actin organisation, at 6 h following seeding on micropatterns, in cells transfected with control siRNA (**a, c**), profilin 1 (Pfn1) siRNA (**b**), mDia1 siRNA (**d, f**) and mDia1&profilin 1 siRNAs (**e**). 20 nM latrunculin A (LatA) was added to control (**c**) or mDia1 knockdown (**f**) cells 10 min after cell attachment. Actin was visualised by phalloidin-staining after fixation. The histograms below each image show the distribution of average RF tilt in the 6–10 μm annulus in cells under corresponding conditions. **g** Average values of RF tilts (mean ±SEM) as a function of the distance of annuli from the cell edge. Histograms and

mean± SEM values (in **a** to **g**) are based on measurements of 194 control siRNA cells, 164 profilin 1 knockdown cells, 86 LatA-treated control siRNA cells, 161 mDia1 knockdown cells, 137 mDia1&profilin 1 double knockdown cells and 70 LatA-treated mDia1 knockdown cells. Scale bar, 10 μm (**a–f**). **h** Western blot showing mDia1 (upper row) and profilin 1 (middle row) protein level in cells treated with scrambled (control), anti-mDia1, anti-profilin 1 or anti-mDia1 plus anti-profilin 1 siRNAs; α-tubulin (bottom row) was used as loading control. Quantification of fold change relative to control was indicated as mean±SD values for 2 experiments. Colour coding in histograms (**b–f**) correspond to those indicated in graph (**g**). See

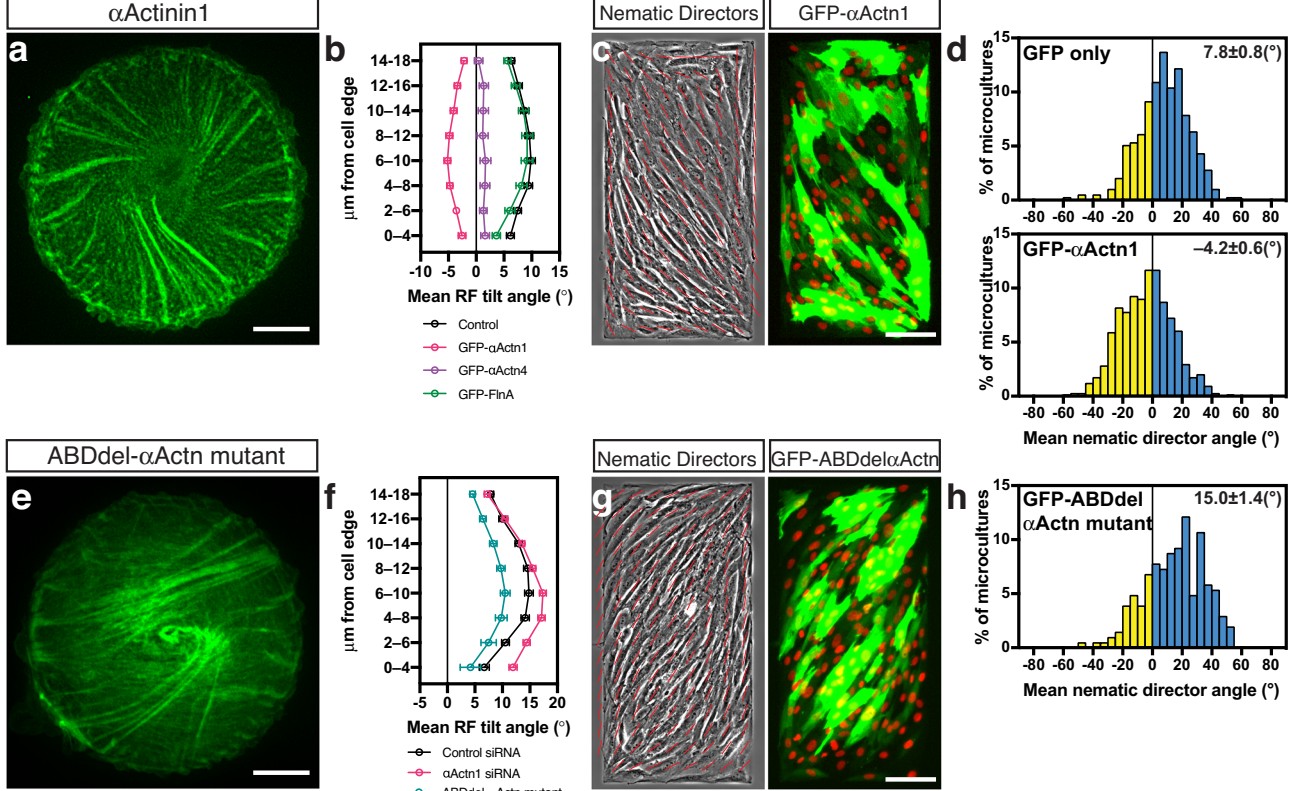

**Fig. 6 | α–Actinin1 overexpression reverses the sign of chirality in individual cells and cell collectives. a** Typical image of GFP-α–actinin1 overexpressing cell with sinistral (reversed) tilt of radial fibres (RFs). **b** Average values of RF tilts (mean ±SEM) as a function of the distance of annuli from the cell edge for control cells (LifeAct-labelled; n = 58), cells overexpressing α-actinin1 (GFP-αActn1; n = 156), α-actinin4 (GFP-αActn4; n = 92) and filamin A (GFP-FlnA; n = 69). Average RF tilts were calculated using images taken during the entire period of observation (12–16 h). **c** Phase-contrast image of rectangular microculture (left) and distribution of GFP-α–actinin1 transfected cells in the same field (right). **d** Histogram showing distribution of the angles of mean nematic directors characterising individual microcultures of GFP-only and GFP-α–actinin1 overexpressing cells (n = 394 and 745 microcultures, respectively) at 48 h following plating. Note that the average nematic directors values are shifted in negative direction in microcultures of GFP-α–actinin1 overexpressing cells as compared to control GFP-only expressing cells. **e–h** Dextral chirality is preserved in cells with suppressed α–actinin crosslinking function. **e** Typical actin organisation in dominant negative GFP-ABDdel-α–actinin

mutant expressing cell visualised by mRuby-LifeAct (pseudo-coloured green). **f** Average values of RF tilts (mean±SEM) as a function of the distance of annuli from the cell edge for control siRNA-transfected cells (n = 203, at 6 h post cell seeding), α–actinin1 siRNA-transfected cells (n = 192, at 6 h post cell seeding) and GFP-ABDdel α–actinin mutant expressing cells (n = 85, imaged for 12–16 h). **g** Phase-contrast image of rectangular microculture (left) and distribution of GFP-ABDdel-α–actinin mutant transfected cells in the same field (right). **h** Histogram showing distribution of the angles of mean nematic directors characterising individual microcultures of GFP-ABDdel-α–actinin mutant overexpressing cells. The histogram was built based on 206 microcultures at 48 h following plating. In (**c, g**) phase contrast images were overlaid with local nematic directors (red lines), nuclei are labelled with Hoechst 33342 (pseudo-coloured red). Negative and positive values in histograms (**d, h**) are coloured in yellow and cyan respectively. Scale bars, 10 μm (**a, e**); 100 μm (**c, g**). See also Supplementary Fig. 9, a–e. For statistical analysis, see Supplementary Table 1, lines 56–81.

anti-clockwise rotation of the nuclei could be seen (Supplementary Movie 7). Remarkably, the mean orientation of the stress fibres deviated from the direction of the long axis of the ellipse in a chiral manner (Fig. 9). Specifically, on average the stress fibres were tilted several degrees (°) to the right relative to the long axis of the ellipses (Fig. 9c and f), forming slashed ellipse Ø configurations (Fig. 9, a and d and Supplementary Fig. 13). An important piece of evidence that the chiral orientation of the stress fibres on elliptical substrate is driven by

chiral tilting of radial fibres, similar to that on circular substrate, was obtained in experiments with latrunculin A treatment. Similar to the situation on a circular pattern, the spreading on an elliptical pattern in the presence of the low-dose latrunculin A resulted in the formation of the system of stress fibres with reversed (sinistral) direction of chirality (Fig. 9b and e, and Supplementary Fig. 13a and b). On average, these stress fibres were tilted several degrees left relative to the long axis of the ellipses (Fig. 9c and f). Moreover, addition of latrunculin A to the

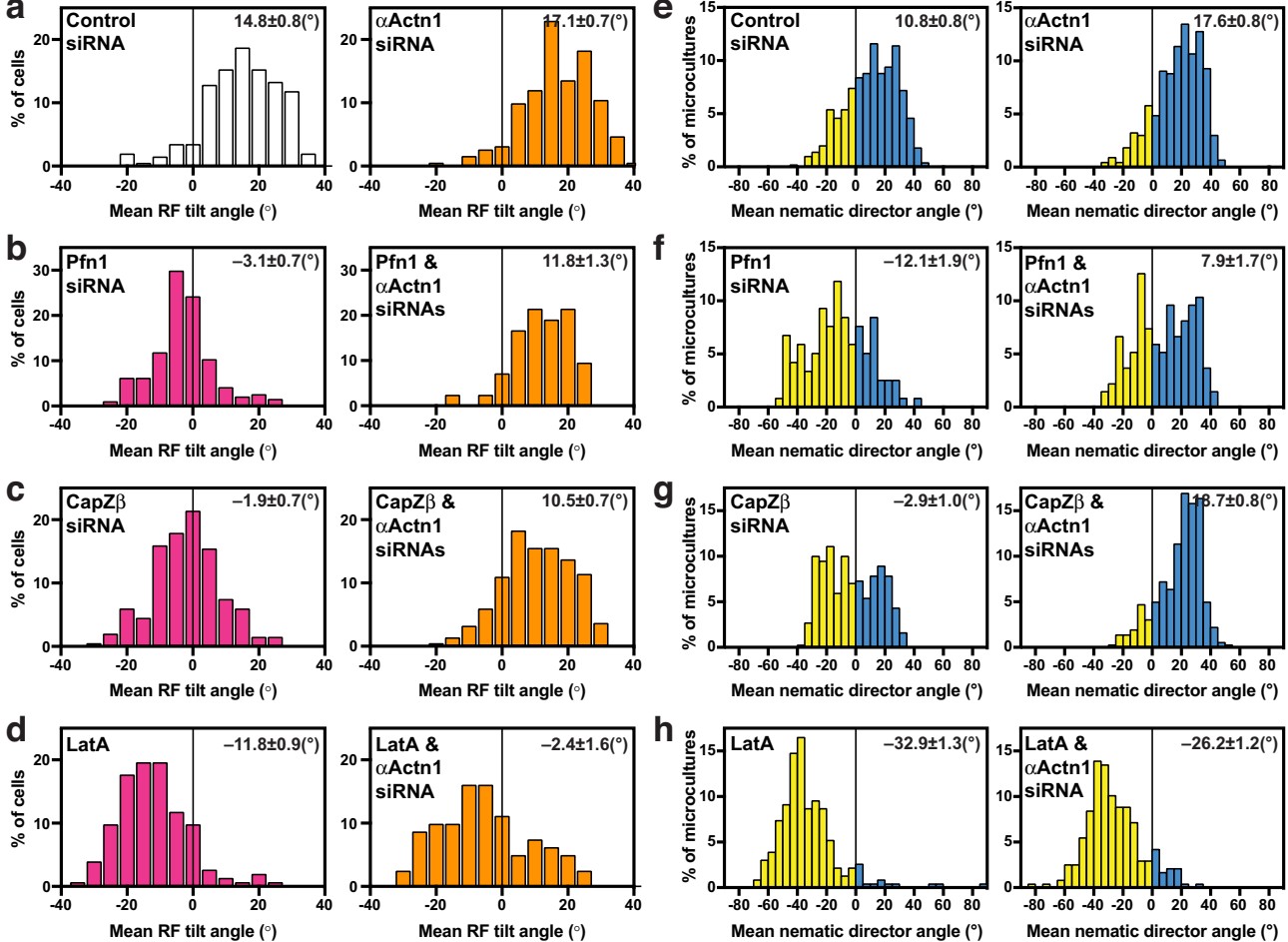

**Fig. 7 | α−Actinin1 is required for the reversal of chirality direction. a-d** The reversal of radial fibre (RF) tilt is α−actinin1-dependent. The histograms show the distribution of average RF tilt in the 6−10 μm (**a**, **b**, **d**) or 8−12 μm (**c**) annuli for (**a**) control siRNA-transfected cells (*n* = 203) and α−actinin1 siRNA-transfected cells (*n* = 192), (**b**) profilin 1(Pfn1) siRNA-transfected cells (*n* = 194) and Pfn1 & α−actinin1 siRNAs-transfected cells (*n* = 42), (**c**) CapZβ siRNA-transfected cells (*n* = 201) and CapZβ & α−actinin1 siRNAs-transfected cells (*n* = 219), and (**d**) 20 nM latrunculin A (LatA)-treated cells (*n* = 153) and α−actinin1 siRNA-transfected cells treated with 20 nM LatA (*n* = 94). Bar colours in histograms (**a**–**d**): white – control cells, magenta – cells treated by agents reversing the chirality direction (Pfn1 siRNA, CapZβ siRNA, LatA), orange – α−actinin1 knockdown cells alone or treated with the chirality reversing agents. See also Supplementary Fig. 9f. **e**–**h** The effect of α−actinin1 knockdown on reversion of the sign of cell alignment angle in microcultures. The histograms showing distributions of the angles of mean nematic directors characterising the microcultures at 48 h following plating for (**e**) control siRNA-transfected cells (*n* = 499) and α−actinin1 siRNA-transfected cells (*n* = 430), (**f**) Pfn1 siRNA-transfected cells (*n* = 118) and Pfn1 & α−actinin1 siRNAs-transfected cells (*n* = 135), (**g**) CapZβ siRNA-transfected cells (*n* = 369) and CapZβ & α−actinin1 siRNAs-transfected cells (*n* = 360), and (**h**) 20 nM LatA-treated cells (*n* = 230) and α−actinin1 siRNA-transfected cells treated with 20 nM LatA (*n* = 237). Negative and positive values in histograms (**e**–**h**) are coloured in yellow and cyan respectively. Mean±SEM values are indicated at the top right corner of each histogram. For statistical analysis, see Supplementary Table 1, lines 82–117.

cells on an elliptical pattern with completely established dextral stress fibres orientation led to re-organisation of the system of stress fibres and development of left-tilted sinistral orientation typical for cells treated with latrunculin A (Fig. 9i to k and Supplementary Movie 8).

## Discussion

The key improvement which made this study possible was the development of rigorous quantitative methods which permitted us to perform a large-scale assessment of the degree of left-right asymmetry in the organisation of the actin cytoskeleton in individual cells and the alignment of cells in confined cell groups. The process of left-right asymmetric actin swirling in isotropic discoid cells is manifested by unilateral tilting of the radial fibres. Thus, using deep-learning computational image analysis, we determined the angles characterising the degree of deviation of these fibres from the radial direction in individual cells. Formation of confluent cell monolayer in microcultures confined to a rectangular micropattern resulted in development of a prevalent angle of cell alignment. We assessed the deviation between

cell alignment axis and the long axis of the rectangle by measuring either the average angle of local nematic directors in phase-contrast images or the average angle of long axes of elliptical cell nuclei. These objective measurements of the left-right asymmetry in individual cells and cell groups allowed us to make quantitative comparisons between the processes of asymmetric actin cytoskeleton organisation and asymmetric cell alignment.

In contrast to earlier studies that focused on a single gene as the main regulator of left-right asymmetry, our study revealed that multiple actin-associated proteins are involved in the control of chiral morphogenesis in individual cells and multicellular microcultures. Among the proteins involved in actin assembly, formins, Arp2/3 complex, cofilins, capping protein and profilin appeared to be potent regulators of left-right asymmetry development. Some of these identified actin regulators were also reported to influence actomyosin-powered cortical flow[35] in *C.elegans* zygote. Both in our system and in *C.elegans*, formins (mDia1 and CYK-1[23] respectively) seem to be important players. This role of formin family proteins might be related

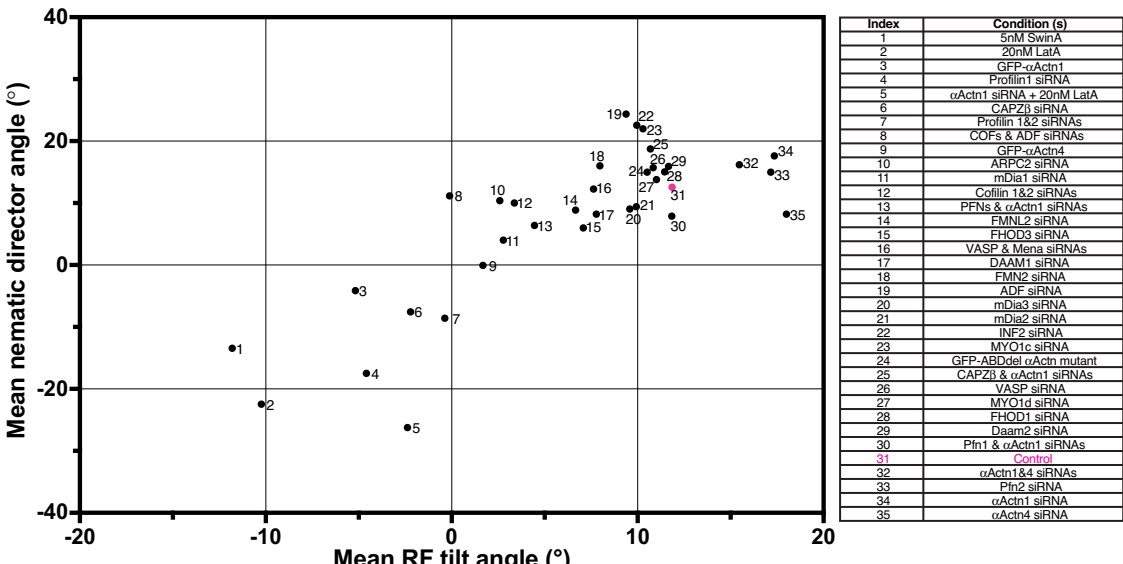

| Index | Condition (s) |
|---|---|
| 1 | 5nM SwinA |
| 2 | 20nM LatA |
| 3 | GFP-αActn1 |
| 4 | Profilin1 siRNA |
| 5 | αActn1 siRNA + 20nM LatA |
| 6 | CAPZβ siRNA |
| 7 | Profilin 1&2 siRNAs |
| 8 | COFs & ADF siRNAs |
| 9 | GFP-αActn4 |
| 10 | ARPC2 siRNA |
| 11 | mDia1 siRNA |
| 12 | Cofilin 1&2 siRNAs |
| 13 | PFNs & αActn1 siRNAs |
| 14 | FMNL2 siRNA |
| 15 | FHOD3 siRNA |
| 16 | VASP & Mena siRNAs |
| 17 | DAAM1 siRNA |
| 18 | FMN2 siRNA |
| 19 | ADF siRNA |
| 20 | mDia3 siRNA |
| 21 | mDia2 siRNA |
| 22 | INF2 siRNA |
| 23 | MYO1c siRNA |
| 24 | GFP-ABDdel αActn1 mutant |
| 25 | CAPZβ & αActn1 siRNAs |
| 26 | VASP siRNA |
| 27 | MYO1d siRNA |
| 28 | FHOD1 siRNA |
| 29 | Daam2 siRNA |
| 30 | Pfn1 & αActn1 siRNAs |
| 31 | Control |
| 32 | αActn1&4 siRNAs |
| 33 | Pfn2 siRNA |
| 34 | αActn1 siRNA |
| 35 | αActn4 siRNA |

**Fig. 8 | The correlation between mean radial fibre tilt angle in individual cells and mean nematic director angle in microcultures.** Mean nematic director angles for rectangular microcultures are plotted against mean radial fibre (RF) tilt angles at the 6–10 μm annulus for each type of treatment. Each dot represents average data from pooled experiments under respective conditions as indicated in the list on the right. The data are ordered and indexed according to ascending radial fibre tilt angles (x-axis). Control cell data point is marked by magenta. Pearson correlation coefficient, r = 0.8104, ****p < 0.0001. Statistical analysis was implemented using GraphPad Prism software. Numbers of cells and microcultures analysed and the values of the means±SEM can be found in Supplementary Table 2. See also Supplementary Fig. 12.

to their rotation at the tip of actin filaments during polymerisation due to helical organization of actin filament[36,37].

A remarkable type of response observed in our study was switching from dextral to sinistral chirality. While mutations reversing the direction of chirality at the organismal level have often been observed (*situs inversus* in vertebrates, reversed chirality of hindgut in flies, and sinistral chirality in snails[1–3]), the reversion of chirality in isolated individual cells was not sufficiently explored. In our studies, we found that the actin cytoskeleton of individual cells can also demonstrate a pattern of organisation that looks like a mirrored reflection of the normal chiral pattern. The most striking examples are knockdowns of profilin 1 (but not profilin 2) and CapZβ subunit of capping protein CapZ, which both led to negative average tilting of the radial fibres in individual cells and clockwise (rather than anti-clockwise) cytoskeleton swirling. Another group of treatments that efficiently reversed chirality direction was treatment with low concentrations of actin polymerisation inhibitors, latrunculin A and swinholide A.

Surprisingly, mDia1, the formin critical for the dextral cell chirality appeared to be dispensable for sinistral chirality induced by the aforementioned treatments. At the same time, knockdown of α−actinin1, which did not interfere with normal dextral chirality in individual cells (ref. [11] and present study), prevented the reversal of direction of individual cells chirality. α−Actinin1 is a major cross-linking protein in radial fibres and its function in chirality determination may depend on its possible role in restricting individual filament rotation and regulation of radial fibre twisting. In our previous paper, we speculated that the torque induced by formin-driven rotation of trapped filament can occasionally be released (when elastic energy of the system approaches some threshold), leading to the rotation of filament in the opposite direction[38]. Such consideration could explain the reversal of cell chirality in α−actinin1 over-expressing cells[11]. In recently published papers[39,40], this model was elaborated and applied to the explanation of the phenomenon of chirality reversal. However, which factor induces filament rotation in cells with sinistral chirality in the absence of mDia1 remains to be elucidated.

Another model connects the direction of cell chirality with the structural organisation of the interactions between radial and transverse actin fibres. We posited previously that the initial breaking of the left-right symmetry starts when the radial actin bundle rotating unidirectionally around its long axis begins to 'roll' on circumferential transverse actomyosin structures in a 'rack-and-pinion' mechanism[11,41]. If the circumferential structures are 'above' the radial actin bundles (along z-axis), then the clockwise-rotation of the bundles (if one looks along the bundle axis from the barbed ends at focal adhesions) results in anti-clockwise swirling in the cell. However, if the circumferential structures are 'below' the radial actin bundles, then the clockwise-rotation of the bundles should produce clockwise swirling. It is possible that some perturbations of the actin dynamics and/or cross-linking could change the mutual position of radial and transverse fibres, thereby changing the direction of cell chirality. Future structural information is needed to evaluate this hypothesis.

Our experimental systems allowed us to perform systematic quantitative comparison between effects of diverse genetic and pharmacological treatments on development of left-right asymmetry in individual cells and multicellular microcultures confined to rectangular patterns. Asymmetric alignment of cells in our microcultures resembles chiral behaviour seen in cells confined to stripes or ring-shaped patterns[8–10]. Our study revealed a remarkable correlation between responses of individual cells and cell collectives in microcultures. With only few exceptions, the treatments which affected formation of asymmetric actin pattern in individual cells also affected asymmetric alignment of cell groups. Treatments that reversed actin chirality direction in individual cells always resulted in a change of the direction of average cell alignment in cell groups. These data, in line with[42], provide strong experimental support to the hypothesis that the development of the chiral organisation in multicellular cultures, tissues, and organs is determined by the chirality of the actin cytoskeleton in the individual cell.

Our data on chiral deviation of the average direction of stress fibres from the long axis of elongated cells confined to an elliptical pattern could shed some light on the relationship between individual and collective cell chirality. Previous study suggests that the

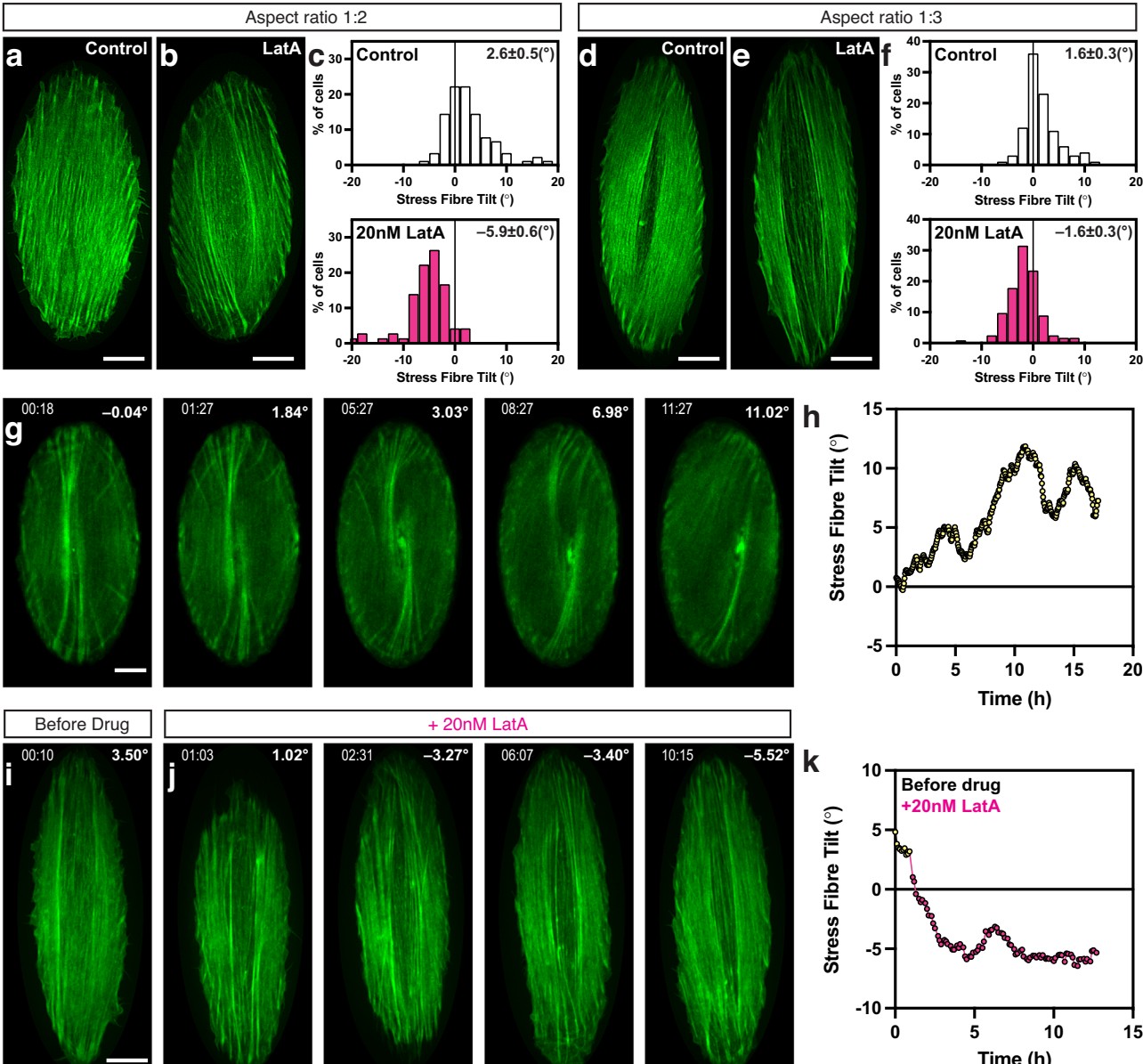

**Fig. 9 | Self-organisation of the actin cytoskeleton in individual cells confined to elliptical micropatterns. a–f** Actin cytoskeleton organisation, visualised by phalloidin-staining, of typical control (**a, d**) and latrunculin A-treated (**b,e**) cells 6 h following plating on elliptical fibronectin-coated patterns with an aspect ratio of 1:2 (**a–c**) and 1:3 (**d–f**) respectively. 20 nM Latrunculin A (LatA) was introduced 10 min after cell plating. The histograms (**c, f**) show the distribution of average tilt of stress fibres relative to the long axis of the ellipses in 100 control and 124 LatA-treated cells (**c**), and 90 control and 72 LatA-treated cells (**f**). White and magenta bars correspond to control and LatA-treated cells, respectively. Mean±SEM values are indicated at the top right corner of each histogram. See also Supplementary Fig. 13. For statistical analysis, see Supplementary Table 1, lines 118–123. **g** A sequence of images of LifeAct-transfected control cell at different time points after plating on an elliptical (1:2 aspect ratio) pattern. See also Supplementary Movie 7. **h** Graph showing the changes in the average stress fibre tilt over time in cell (**g**). The graph was smoothed by averaging 4 sequential measurements. **i, j** The actin stress fibres in LifeAct-transfected cell on an elliptical (1:3 aspect ratio) pattern before (**i**) and at different time points after the addition of latrunculin A (**j**). See also Supplementary Movie 8. **k** Graph showing the changes in the average stress fibre tilt over time in cell (**i–j**). (**g, i, j**) The time points (hh:mm) and corresponding stress fibre tilt (°) are indicated in upper left and right corners of the images, respectively. For display and analysis purpose, all images of cells on elliptical pattern were placed on black background and aligned along vertical direction. Scale bars, 10 µm (**a, b, d, e, g, i**). See also Supplementary Fig. 13.

orientation of an individual cell starts with orientation of the systems of focal adhesions and stress fibres, followed by orientation of the cell as a whole[43]. We postulate that cells could preserve intrinsic actin cytoskeletal chirality when aligned with the boundaries of the adhesive pattern, in such a way that aligned cell migrates preferentially anti-clockwise around the pattern's boundary. In this scenario, individual cell anti-clockwise bias in collectives might causes an anticlockwise biased streaming of cells around the boundary of the rectangular micropattern because of the combination of contact inhibition of

locomotion, lateral interactions and leader-follower behaviours ultimately aligning cells' polarisation axes with each other to collectively evolve into the ⋈-orientation. This biased streaming of cells along the boundaries was also observed in other independent studies on collective cell chirality on stripes and rings patterns[9,10]. The future goal is to elucidate how chiral actin organisation drives chiral cell movement along the boundary of adhesive islands.

The hypothesis of the role of asymmetric self-organisation of the actin cytoskeleton in the emergence of chirality in tissues, organs, and

even in whole organism could unify the existing data on the development of left-right asymmetry in snails[20,21] and in *Drosophila*[5,6,19,22], and probably for asymmetric heart looping in vertebrate development[4]. The formation of asymmetry of heart and visceral organs positioning in vertebrate deserves special discussion. It is well-established that in many species the key asymmetric factor triggering the signalling cascade determining left-right body asymmetry is an asymmetric flow generated by ciliated cells in the node (left-right organiser of the embryo)[1,2]. Recent studies have shown, however, that in birds and reptilia the nodal asymmetry does not depend on cilia[1]. Thus, the cilia may function as an amplifier, but not the primary source of asymmetry. Since the position and orientation of basal bodies can in principle be regulated by the actin cytoskeleton[44], this suggests that the primary asymmetric factor in nodal cilia-dependent systems could still be the intrinsic asymmetry of the actin cytoskeleton. Extensive future studies are necessary to explore this possibility. In conclusion, our study revealed an actin polymerisation-dependent mechanism of establishment of left-right asymmetry in individual cells and cell groups which could be involved in the development of left-right asymmetry in organs and organisms.

## Methods

### Cells and plasmids transfection

Human foreskin fibroblasts (HFF) from American Type Culture Collection (catalog no. SCRC-1041) were cultured in Dulbecco's modified Eagle's medium high glucose supplemented with 10% fetal bovine serum (FBS), 1 mM sodium pyruvate and antibiotics (penicillin and streptomycin) at 5% $CO_2$ at 37 °C. Enucleated cells were generated as described in our earlier work in ref. [12]. Briefly, non-transfected or LifeAct-transfected cells were seeded onto plasma-treated plastic coverslips (ibidi GmbH) and left overnight to ~70% confluency. Next, cells were treated with 5 ml of 10 µg ml$^{-1}$ cytochalasin B (Sigma) in a 50 ml falcon tube and centrifuged at 10,864 g (Beckman centrifuge X30R) for 1 h at 37 °C to enucleate cells. Cells were washed three times with complete medium and allowed to recover for at least 2 h in complete medium following which cells were trypsinised for seeding onto the micropatterned substrate for the experiment. Presence of nuclei were verified by labelling with Hoechst 33342 (10 µg ml$^{-1}$ for 10 min) and live imaging of the nucleus. Cells were transfected with DNA plasmids via electroporation (Neon® transfection system, Life Technologies) following manufacturer's instructions. Electroporation condition consists of two pulses of 1150 V for 30 milliseconds. Expression vectors encoding the following fluorescent fusion proteins were used: LifeAct-GFP[11], mRuby-LifeAct[12], mEmerald-mDia1-C-14 (Addgene plasmid # 54156), pEGFP-C1 (Clontech), GFP-α−actinin1[11], GFP-α−actinin4 (gift of Dr. M. Pan, Mechanobiology Institute, Singapore), GFP-ABDdel-α−actinin1[11] (gift of Dr P. Roca-Cusachs, University of Barcelona, Barcelona, Spain), EGFP-Filamin A (gift of Dr. M. Sheetz, Mechanobiology Institute, Singapore), mCherry-Cofilin 1 (gift of Dr. C. G. Koh, Nanyang Technological University, Singapore), mCherry-Profilin1 (Addgene plasmid #55121), Pfn1-P2A-eGFP (Clone ID: OHu24169; GenScript USA Inc) and GFP-VASP[11]. Fluorescence-activated cell sorting (FACS) were performed to select for upper 30% of cells expressing high level of fluorescent fusion proteins of pEGFP-C1, GFP-α−actinin1, GFP-α−actinin4 or GFP-ABDdel-α−actinin1. All cell culture and transfection reagents were obtained from Invitrogen. Other chemicals and reagents were obtained from Sigma, unless otherwise stated.

### siRNA transfection

Cells were seeded into a 35 mm dish on day 0 and transfected with 100 µM of siRNA using Lipofectamine RNAiMAX on days 1 and 2. For experiment involving individual cells, siRNA-transfected cells were trypsinised on day 4 and replated onto circular micropatterns. For experiment involving cell microcultures, siRNA-transfected cells were trypsinised on day 3 and replated onto rectangular micropatterns. As needed, transfection of plasmids via electroporation into siRNA-treated cells were performed on day 3 and cells were replated on day 4. siRNA transfected cells had their proteins or RNAs extracted on day 4 for immunoblotting or RNA sequencing respectively. siRNAs used in this study are listed in Supplementary Table 3.

### Micropatterning of substrates

Cells were seeded on substrates containing either 1,800 µm$^2$ circular or elliptical micropatterns of different aspect ratio (individual cell experiment), or 300 × 600 µm rectangles (multicellular microculture experiment). Each micropatterned substrate was fabricated by stencil patterning as previously described in our earlier work[12]. Briefly, polydimethylsiloxane (PDMS) (Sylgard 184 kit, Dow Corning) was cast on the photoresist mould, containing micropattern designs of interest, using a 10:1 ratio (w/w) of elastomer to crosslinker and cured for 2 h at 80 °C. The crosslinked PDMS layer was peeled off and stamps were cut out manually. The PDMS stamp was then inverted and placed onto a hydrophobic uncoated 35 mm µ-dish (ibidi GmbH). Norland Optical Adhesive 73 (NOA-73, Norland Inc.) was deposited along an edge of the stamp and allowed to flow through the gaps between the PDMS stamp and dish by capillary action, upon which the stamp was sealed on all sides using NOA-73. The NOA-73 stencil was cured under ultraviolet illumination for 15 s. After peeling off the PDMS stamp, the stencil and dish were incubated with fibronectin (Calbiochem, Merck Millipore) at a concentration of 50 µg ml$^{-1}$ in 1 × PBS at 4 °C overnight after a brief degassing at 10 mbar. At the end of the incubation, the fibronectin solution was aspirated, and the stencil was removed. The printed dish bottom was passivated with 0.2% Pluronic acid-$H_2O$ for 10 min. Finally, the passivated dishes were washed thrice with 1 × PBS before cell seeding.

### Assessment of individual cells on circular micropattern

Cells were seeded on printed dishes containing circular micropatterns at a density of $5 \times 10^4$ cells ml$^{-1}$ for 10 min. The medium containing unattached cells was then replaced with fresh DMEM. After 6 h incubation, the cells were fixed using 4% paraformaldehyde (Tousimis, USA) in PBS for 10 min, followed by three 1 × PBS washes. Cells were permeabilised using 0.1% Triton-X-100 in PBS, and then blocked with 2% bovine serum albumin (BSA)-PBS for 1 h at room temperature (RT) before incubation with appropriate labelling reagents. Actin and nucleus staining were performed using phalloidin (Molecular Probes) and Hoechst 33342 (Invitrogen), respectively. For live cell imaging experiment, cells were seeded on circular micropatterns at a density of $5 \times 10^4$ cells ml$^{-1}$ for 10 min. The medium containing unattached cells was then replaced with Leibovitz's L-15 containing 10% FBS. Cells were left for at least 2 h before imaging at 37 °C with 5% $CO_2$. Time-lapse images for 12−16 h at 10−20 min intervals and Z-stacks of step-size 0.35 µm with total height of 10−15 µm were acquired with a spinning disc confocal microscope (PerkinElmer Ultraview VoX) attached to an Olympus IX81 inverted microscope, equipped with a 100× oil immersion objective (1.40 NA, UPlanSApo), an EMCCD camera (C9100-13, Hamamatsu Photonics) for image acquisition, and Volocity software (PerkinElmer) to control the set-up. Fixed samples were also imaged with the same step-up. Maximum projection of the Z-stack images was performed with Volocity software or with Fiji software and exported as 16-bit TIFF files (512 × 512 pixel and 0.138502 µm pixel$^{-1}$). Each image contained a single cell and these images were subsequently used for deep learning-based identification of radial fibres.

### Assessment of individual cells on elliptical micropattern

Cells were seeded on printed dishes containing elliptical micropatterns with an aspect ratio of 1:2 (34:68 µm) or 1:3 (27.5:84 µm) at a density of $5 \times 10^4$ cells ml$^{-1}$ for 10 min. The medium containing unattached cells was then replaced with fresh DMEM. After 6 h incubation,

the cells were fixed using 4% paraformaldehyde (Tousimis, USA) in PBS for 10 min, followed by three 1×PBS washes. Cells were permeabilised using 0.5% Triton-X-100 in PBS, and then blocked with 5% bovine serum albumin (BSA)-PBS for 1 h at RT before overnight incubation at 4 °C with appropriate primary antibodies. Cells were then incubated for 45 min at RT with appropriate labelling reagents: AlexaFluor-conjugated secondary antibodies (Molecular Probes, dilution 1:500) and Phalloidin (Molecular Probes). For live cell imaging experiment, cell seeding was as above except that the medium containing unattached cells was then replaced with Leibovitz's L-15 containing 10% FBS. Time-lapse images at 3 min intervals and Z-stacks of step-size 0.3 μm with a total height of 11 μm were acquired using a 60× oil immersion objective (1.35 NA, UPlanSApo) with the spinning disc confocal microscope (PerkinElmer Ultraview VoX). Fixed samples were also imaged with the above step-up using a 100× oil immersion objective (1.40 NA, UPlanSApo), or on the spinning disc confocal microscopy coupled with the live super-resolution (SR) module (Roper Scientific) attached to a Nikon Eclipse Ti-E inverted microscope with Perfect Focus System, equipped with 100× oil immersion objective (1.4 NA, PL APO VC), a sCMOS camera (Photometrics Prime 95B) for image acquisition, and MetaMorph software (Molecular Devices) to control the set-up. Maximum projection of the Z-stack images was performed with Volocity software or with Fiji software and exported as 16-bit TIFF files. Angle (°) of stress fibre tilt was measured using images of actin cytoskeleton in cells on elliptical micropattern labelled either by phalloidin or LifeAct with the OrientationJ plugin in Fiji software. As needed, image was rotated using transform tool with bicubic interpolation to align the long axis of the elliptical cell to the vertical orientation. An elliptical mask was then applied to the image and the angle of stress fibre tilt of each elliptical cell, single or in a time-lapse series, was measured using the OrientationJ Measure plugin.

## Assessment of microcultures on rectangular micropatterns

Cells were seeded on rectangular micropatterns at a density of $1 \times 10^5$ cells ml$^{-1}$ for 20 min. The medium containing unattached cells was then replaced with fresh DMEM and cell microcultures were incubated for a total of 48 h before cell fixation using 4% paraformaldehyde-PBS for 10 min. Just prior to fixation, cell nuclei were stained with 1 μg ml$^{-1}$ Hoechst 33342 for 10 min. Phase-contrast and fluorescence images of cell microcultures were taken using a 20× air objective (0.45 NA, LUCPLFLN20X, Olympus) on an Olympus IX81 inverted microscope, equipped with Andor Neo 5.5 sCMOS camera and light source (Lumencor SOLA SE Light Engine). Single plane images of phase contrast and DAPI channels were taken. Each image contained a single rectangular cell microculture and these images were subsequently used for measurement of average nematic directors angle and nuclei orientation angles in rectangular microcultures.

## Drug treatment

For drug inhibition studies, 10 min following cell seeding on micropatterns, the medium containing unattached cells was replaced with fresh medium containing either 20 nM latrunculin A (Santa Cruz Biotechnology Inc., SCB Inc.) or 5 nM swinholide A (SCB Inc.). For experiments that lasted more than 24 h, fresh drugs were added every 24 h until the end of the observation period. All inhibitors remained in the medium during the entire period of observation, except in drug washout experiments.

## Segmentation of radial fibres

Images of the actin cytoskeleton were first converted to 8-bit and the 'Enhance brightness/contrast' function in Fiji software was used with the 'saturated pixels' parameter set to the default of 0.35. A Unet-ResNet50 deep learning model[45], implemented in Python, was trained to identify actin radial fibres in cells confined on circular micropattern. Briefly, the following steps were taken. The model was trained using 32

images of actin cytoskeleton labelled by fluorescent protein tagged-LifeAct. These training images comprise of cells with their actin cytoskeleton in a radial or chiral organisation, and images of varied intensities were selected. Data augmentation was done using the Albumentations library[46]. The ground-truths (binary, 8-bit) were prepared by manual demarcation of actin radial fibres in Fiji. The code and complete list of parameters of the trained deep learning model is available via https://github.com/gohweijia/Cell-Chirality-Analysis. This trained model was used to identify radial fibres in both phalloidin- and LifeAct-labelled cells, returning a 32-bit image of identified radial fibres. Segmentation of these identified radial fibres was performed using a custom MATLAB script, in which background subtraction (with rolling ball of 30-pixel radius) and then Niblack local thresholding[47], with window size of 15 × 15, k = −0.3 and offset = −0.01, were applied. The resulting binary image was then skeletonised using the MATLAB built-in function, bwmorph. Intersecting radial fibres were separated by branch point removal, and fibre segments with similar orientation (angle difference ≤ 30°) and at nearby position (distance ≤ 30 pixels) were connected as a single fibre.

## Measurement of radial fibre tilt angles

The following procedures were performed using a custom MATLAB script unless otherwise stated. Cell segmentation was performed by thresholding the Gaussian-smoothed (sigma value set to 3) actin image using Otsu binarization (threshold value set to 0.4 of Otsu auto threshold), followed by a series of mathematical morphological operations (imclose, imfill, imerode). Cell centroid and cell spread area were calculated based on this cell mask. The cell mask was also used for generating concentric ring masks of 4 μm in width starting from the cell edge, with 2 μm increments, for masking the segmented radial fibres. In each ring, the angle of each radial fibre segment was measured relative to the cell edge. See also Supplementary Fig. 1c. The angle at the cell edge $\theta$, computed using Python, was given by the formula $\theta = \arcsin(\frac{r \sin \theta_r}{R})$, where $\vec{R}$ connects the cell centroid and intersection of the continuation of the radial fibre segment with the edge of the cell and $\vec{r}$ connects the cell centroid and intersection of radial fibre with outer edge of the annulus. $\theta$ and $\theta_r$ are the angles between the radial fiber segment and $\vec{R}$ and $\vec{r}$ respectively (Supplementary Fig. 1c). Based on visual inspection, actin fibre segments with $\theta_r$ more than or equals to 68° were unlikely to be radial fibres and were omitted from the analysis. The average inflation of area of the cell mask relative to the area of the micropatterns (1800 μm$^2$) was estimated to be 63.353 μm$^2$ using a dataset of ~100 cells. This constant was subtracted from the cell area before the computation of cell radius R. Only cells with area between 1700 and 2000 μm$^2$ were analysed.

## Computation of nematic directors and nuclei orientation

The following procedures were performed using a custom MATLAB script unless otherwise stated. First, identification and segmentation of individual rectangular microculture using phase contrast images was done by performing a Wiener filter with a neighbourhood size of 20 × 20 pixels to remove image noise. This was followed by an entropy filter with a 3 × 3 pixels structural element and morphological opening with a 9 × 9 pixels structural element. This results in an image that differentiates between areas with and without cells. Otsu binarization was then performed to segment the image, the segmented area at the centre of the image was selected as the segmentation mask. This serves as an indicator of the area covered with cells. The bounding box enclosing this segmented area serves to represent the dimensions of the microculture. Only microcultures with bounding box width of 225 to 375 μm and height of more than 550 μm, and with a segmentation mask that covered more than 80% of the bounding box area were analysed. For each bounding box, the centre 200 × 500 μm region of interest was used for subsequent steps in the measurements of average nuclei orientation and average nematic director orientation.

Segmentation of the nuclei was achieved using NICK adaptive binarization[48,49]. A Wiener filter using a neighbourhood size of $9 \times 9$ pixels was performed prior to segmentation. The concaved-point based splitting algorithm[50] was used to separate any overlapping nuclei. Segmented objects above the size of 10000 pixels were removed as these corresponded to the background regions, while objects smaller than 500 pixels were also removed as these were either fragmented nuclei or noise regions that were segmented by chance. In addition, only nucleus that had a centroid position that laid within the bounding box of the segmented phase contrast image was selected for further analysis. The number of nuclei in the bounding box was also counted. Microcultures that had less than 50 nuclei were removed as these microcultures often had too few cells to cover the entire rectangular micropattern. The orientation of these nuclei was then calculated based on the angle of the long axis of a fitted ellipse with respect to the long axis of the rectangular micropattern. Alignment of the cell group was determined based on the mean resultant length[51] of the nuclei orientation. A cutoff value of 0.35 was selected and any rectangle with a mean resultant length greater than that was classified as aligned. The mean nuclei orientation per aligned microculture was determined by calculating the mean of all the orientations of the individual nuclei within a single microculture.

Local cell orientation in the phase contrast image was calculated by obtaining the nematic director field as described in ref. [26]. Briefly the orientation tensor was obtained using OrientationJ implemented in Fiji and the nematic director was obtained using a window size of $60 \times 60 \, \mu m^2$ and 70% overlap. The orientation of each directors was measured as the angle relative to the long axis of the rectangular micropattern. The orientation of the directors within the region of interest was then used to determine the alignment of the microculture in a similar manner as that for the nuclei orientation. A higher cutoff value of 0.5 for alignment was set due to more coherent nature of the nematic directors. The mean nematic director angle per aligned microculture was determined by calculating the mean of all the orientations of the directors within a single microculture.

## Immunoblotting

Cell pellets were collected in RIPA buffer (SCB Inc.) supplemented with $2 \, \mu L \, ml^{-1}$ protease inhibitors cocktail (Sigma, catalogue no. P8340), and were then mechanically lysed by syringing through a 27.25 G needle on ice. Protein concentration was quantified using the Micro BCA Protein Assay Kit (Thermo Scientific) according to manufacturer's instructions. 20 μg of protein lysate was dissolved in 1×Laemmli sample buffer supplemented with 10% 2-mercaptoethanol, and separated by 4–20% SDS-polyacrylamide gel (GenScript USA Inc) electrophoresis at 100 V for 1 h and then transferred to a 0.4 μm pore size PVDF membrane (Thermo Scientific, catalog number 88518) at 100 V for 2 h for formin proteins and 1 h for other proteins in an ice bath. The PVDF membrane was blocked using Intercept® (TBS) Blocking Buffer (LI-COR, Inc.) or 5% nonfat milk (Bio-Rad) in Tris-buffered saline with 0.1% Tween 20 (TBS-T) for 1 h at RT before incubation at 4 °C overnight with appropriate primary antibodies. Primary antibodies were diluted in Intercept® (TBS) Blocking Buffer containing 0.1% Tween-20 at their respective concentrations summarised in Supplementary Table 4. After washes in TBS-T, the membrane was probed with either IRDye® 680RD Goat anti-Rabbit IgG (LI-COR, dilution 1:5,000) or IRDye® 800CW Goat anti-Mouse IgG (LI-COR, dilution 1:15,000) for 1 h at RT. The membrane was then washed in TBS-T before fluorescent detection with an Odyssey® CLx imaging system at a resolution of 169 μm and 'medium' quality settings on Image Studio software. Alternatively, the primary antibody binding was processed for ECL detection with appropriate HRP-conjugated secondary antibodies (Santa Cruz Biotechnology, catalogue no. sc-2004/5, dilution 1:10,000) and acquisition using Image Lab Touch Software on GelDoc Go Imaging System (Bio-Rad). Protein ladder used include: Precision Plus Protein™ Kaleidoscope™ Prestained Protein Standards (Bio-Rad; catalog number 1610375), Broad Multi Color Pre-Stained Protein Standard (GenScript USA Inc; catalog number M00624) and Cruz Marker™ Molecular Weight Standards (SCB Inc; catalog number sc-2035).

## Transcriptome profiling by RNA sequencing

RNAs were extracted using RNeasy® Plus Universal Kits (Qiagen) according to manufacturer's instructions. Library was prepared using TruSeq Stranded mRNA LT Sample Prep Kit and sequenced using NovaSeq6000 Illumina platform. Alignment was performed (STAR aligner) and trimmed reads were mapped to GRCh38 reference genome (BioProject: PRJNA312570) with HISAT2, splice-aware aligner. Gene expression was presented using transcript per million (TPM) reads.

## Statistics and reproducibility

The numbers of samples (n) of individual cells and microcultures analysed for all of the quantitative data are specified in the figure legends and summarised in Supplementary Table 2. No statistical method was used to predetermine sample size. All images are representative of at least three independent experiments, except for Fig. 3c, Fig. 5e,f, Fig. 6, Fig. 9i,j and Supplementary Figs. 8–10, which were from two independent experiments and Supplementary Fig. 3 was from a single experiment. The quantified immunoblots in Supplementary Figs. 11 and 7c were from three independent experiment and Fig. 5 was from two independent experiments. The rest of the quantified immunoblots were obtained in a single experiment. Transcriptome profiling of gene expression levels by RNA-sequencing (RNA-seq) shown was from a single experiment, except for Supplementary Figs. 2a and 11a (mDia1 and Profilin 1 knockdown cells) which were from two experiment. All supplementary videos show representative data from at least two independent experiments. Prism software (version 9.4.1; GraphPad Software, LLC.) was used for data and statistical analysis, including frequency distribution for histograms plot, Mann-Whitney test for comparison and reporting of significant difference between two groups, Kruskal-Wallis test for multiple comparisons across groups and reporting of any significant difference between groups, Wilcoxon signed-rank test to estimate the difference of median values of samples from zero, and sum of 2 Gaussian fit. For frequency distribution, a bin width of 5° is used for all histograms, except for histograms representing stress fibre tilt which uses a bin width of 2°. Statistical significance is defined as $P < 0.05$. Mann-Whitney test was two-tailed. Kruskal-Wallis test was implemented with Dunn's multiple comparisons test and report multiplicity adjusted P value for each comparison. Wilcoxon signed-rank test was implemented against a hypothetical value of zero. For each histogram, we compared the fit of Gaussian versus Sum of two Gaussians with default software recommended parameters, including least squares regression, asymmetrical confidence intervals (CI) using 95% confidence level and plot 95% confidence bands. There was neither special handling of outliers nor weighting. Bimodal distribution is defined by the following parameters: preferred fit by the Sum of two Gaussians and represented by 1 negative (Mean1) and 1 positive (Mean2) means demonstrating chiral sign in opposite directions. Reliability of bimodality is checked by plotting the fitting curve with 95% confidence bands over the histogram (see Supplementary Fig. 4a).

## Reporting summary

Further information on research design is available in the Nature Portfolio Reporting Summary linked to this article.

# Data availability

All data generated or analysed during this study are included in this published article (and its Supplementary Information files). Raw datasets corresponding to all histograms in the main figures,

uncropped western blots and gene expression profiles presented as transcript per million (TPM) reads are provided as a Source Data file with this paper. The raw dataset for all other graphs presented in this study are available from the corresponding authors on reasonable request. Source data are provided with this paper.

## Code availability

Custom-written code used to analyse the data in the current study is available from the corresponding authors on reasonable request. The image analysis code for radial fibre segmentation and measurement of their tilt angle can be found at Github - https://github.com/gohweijia/Cell-Chirality-Analysis.

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

## Acknowledgements

We thank M. M. Kozlov (Tel Aviv University, Israel) and T. Hiraiwa (MBI, Singapore) for discussion, T. B. Saw (MBI, Singapore) for consulting on nematic cell orientation, M. Davidson fluorescence protein collection (The Florida State University, Tallahassee, USA), P. Roca-Cusachs, M. Pan, M. Sheetz and C. G. Koh for providing reagents, A. Wong (MBI, Singapore) for expert help in paper editing, P. Kathirvel (MBI, Singapore) and H. Chen (MBI, Singapore) for expert help in molecular work and FACS, and the SIMBA microscopy facility and nanofabrication core facility at the Mechanobiology Institute for technical help. The research is supported in part by the Singapore Ministry of Education Academic Research Fund Tier 2 (MOE Grant No: MOE2018-T2-2-138, awarded to A.D.B; MOE2019-T2-1-099 and MOE2019-T2-02-014; awarded to P.K.), and Tier 3 (MOE Grant No: MOE2016-T3-1-002 and MOET32021-0003; awarded to A.D.B), the National Research Foundation, Prime Minister's Office, Singapore, and the Ministry of Education under the Research Centers of Excellence program through the Mechanobiology Institute, Singapore (ref no. R-714-006-006-271), and by the Singapore Ministry of Health's National Medical Research Council under its Open Fund - Young Individual Research Grant (Grant No: OFYIRG18may-0041; awarded to Y.H.T).

## Author contributions

Y.H.T. and A.D.B conceived and designed the experiments. Y.H.T, W.J.G and X.Y. performed most experiments. J.H., I.Y.Y.T., S.S., S.J., S.F.H.B., P.K., W.H., J.Y., Y.A.B.L., and V.T. contributed to some experiments. W.J.G, X.Y. and H.T.O. developed image analysis tools. A.M. contributed to data analysis and theoretical considerations in Discussion. Y.H.T., A.M., and A.D.B. wrote the manuscript with input from all of the authors.

## Competing interests

The authors declare no competing interests.
