## [Peer Review File · Nature Communications]

REVIEWER COMMENTS

Reviewer #1 (Remarks to the Author):

Cell and tissue-scale chirality, which results somehow from the scaling-up of molecular- and polymer-scale chirality, is fascinating and fundamental to living things. Tee, Bershadsky and colleagues had previously demonstrated a key role for the crosslinker alpha-actinin in single-cell chirality and provided a conceptual model of the accumulation of torsional stresses within actin networks generated by membrane-anchored formins and stored via actin crosslinking. Here, they further demonstrate molecular mechanism for the existence and chirality of the tilt of radial actin bundles in single cells cultured on circular patches. The movies of this process are very helpful and visually stunning.

The work is highly quantitative, careful and thorough. For example, the demonstration of each protein depletion's effect on other proteins of interest in addition to the target is much appreciated. That said, limited additional quantitation of their existing data with newly developed computer vision would add key insights into mechanism. In addition, the current version of the manuscript suffers from inadequate discussion of physical mechanism. I make some suggestions here for how the authors could fill out their discussion – I look forward to learning what they think.

Major points:

1) Radial actin bundles appear anchored to the nucleus, which itself swirls around the cell. What structure provides the pulling or pushing force for these various movements? Is generation of F-actin, spooling out from peripheral formins, sufficient? Or is the nucleus drawn through the cytoplasm by something else, taking the radial bundles with it? I doubt this latter idea, since in one of the two examples shown, the nucleus has a much more limited travel than the radial bundle apexes.

2) I don't agree that mDia1 depletion "abolished the chirality of microculture orientation" since there are still many micropatches with normal chirality, and there are many with strong but reversed chirality. The proportion of micropatches with no chirality is unchanged. When the distribution becomes multimodal, the mean angle is not particularly meaningful – the center of each peak should instead be reported.

3) How do groups of cells acquire similar tilt? Are they pushing on each other or pulling? The authors' computer vision should be applied to their wonderful timelapse image series of these patches' evolution. Do the pointed-into corners (upper-right and lower-left in controls) lead the way, or do the upper-left and lower-right corners first get cleared out instead? Does the theory in Reference 9 make any predictions that the authors can test?

4) The authors should connect their work here to the examination of the separate contributions of torque generation (formin-based polymerization, myosin-based translocation) versus torque storage (crosslinkers, membrane anchoring)? Their candidate list contains both.

5) Related: The similar effect of depleting formins and the Arp2/3 complex is puzzling. The authors say that the proteins whose depletion phenocopies formin depletion "function in cooperation or competition with formins" - why would these two opposing activities have the same effect? Some discussion should be provided. None of the treatments should change the chirality of F-actin, so how is cellular chirality reversed? Could Arp2/3 depletion and other treatments that attenuate actin polymerization affect cell chirality by changing the degree to which the formin stays associated with the barbed end, thus changing its ability to store torque into the filament. If the formin slipped/swiveled as it added actin

subunits, torque would not be stored in the filament. Perhaps there is a formin-slippage regime wherein it essentially slips the opposite direction, building torque of the opposite handedness into the filaments. I'm having a hard time describing this, but it would be similar to the phenomenon of how a car's hubcap can appear to turn backwards with respect to the wheel's revolution, based on the framerate of a camera or your eye being just shorter than the revolution rate. Perhaps the direction and magnitude of torque on F-actin could be tested by assessing relative susceptibility to cofilin severing, or perhaps F-actin probes (CH domains, LifeAct, etc) have different affinities to super-coiled vs under-pitched F-actin (I am taking the liberty to invent terms for actin that has more or less torque.) The authors should include substantial discussion of how their perturbations can cause cellular chirality reversal; I'm keen to know what they think of these suggestions.

Minor points:

1) I'm not sure it's accurate to say that "the majority of multicellular organisms demonstrate approximate bilateral symmetry" since that is not true for any land plants or fungi that I know of, or many animals including echinoderms, some gastropods, and probably many more of which I am less aware. You could say "many animals demonstrate approximate bilateral symmetry."

2) Schonegg, Hyman and Wood needs to be referenced with 11-14, as they were the first to show what is attributed to reference 12 here.

Reviewer #2 (Remarks to the Author):

In this manuscript Tee et al use an assay previously developed by this group in which fibroblasts are seeded on micro patterned circular surfaces to systematically study the role of actin regulators in the emergence and handedness of cellular chirality. In addition, they have extended their findings to the chirality of cell collectives on rectangular surfaces. They found further evidence that supports the hypothesis that chirality of cell collectives arises as a result of cell-intrinsic chirality. Overall the work is exciting, provides novel insights into the molecular control of cell chirality, and makes the perhaps central claim that cellular chirality emerges in nontrivial ways from molecular constituents, and as such can be tuned in several ways. The conclusions are solid and backed up by experimental results (with some exceptions, see below). This paper should be published once the main concerns below are addressed.

Main concerns:

- Did the authors rule out that the image analysis pipeline used to segment radial actin fibers introduces bias? For example, if an RNAi decreases bundle width or curvature, is it segmented better/worse? Does the method miss bundles if their intensity drops in a given RNAi condition? Or vice versa, does the method pick up more bundles if the background level of F-actin is reduced?

- Some RNAi's, for example profilin1, cofilin1/2 and CapZ, result in only small reduction in protein levels, as judged by eye from the western blots, i.e. the ratio between loading control and protein of interest. On top of this, the RNAi effect on cell chirality of profilin1 KO alone and CapZ KO alone seems, by eye, weaker in figure 3 (J and K respectively) than in figure 2 (B and E respectively). Actually, by eye the distribution of angles seems around zero, e.g. no chirality, in figure 3. The authors should verify with statistical tests whether these distributions are different from zero. Similarly, authors should address whether the

distributions of profilin1 RNAi in fig 2 and 3, and CapZ RNAi in figure 2 and 3 are statistically the same or different.

- Related to previous point: If the distributions of profilin1 and capZ in figure 2 and 3 are different, this would suggest that the knockdown efficiency is different in different experiments. Can the authors comment on this? If true, this would raise a couple of additional questions: 1) Cofilin1/2 knockdown decreases, but does not reverse chirality. This might be due to poor knockdown efficiency at the protein level. 2) The double knockdowns in fig 3 (for example profilin1 with alpha actinin knockdown) might have different knockdown efficiency than either single knockdown, which could also explain the results in case of profilin1+alphaAct1 (e.g. no difference from control because of low knockdown efficiency).

- In case of cofilin1/2 KO there appears to be a 1.5 fold up regulation of alpha-actinin (fig S10). Does this affect the conclusions drawn on the effect of cofilin1/2?

- The finding that profilin1, but not Formin knockdown, results in chirality reversals is very interesting. According to the author's previous explanation for chirality reversals (Tee et al 2015), both clockwise and counter-clockwise chirality depends on Formins. In one case the authors hypothesised this is due to the 'stair-stepping' mode of Formin while in the other it is due to 'screw-stepping' mode of Formin. It would be very informative to the field if the authors could verify by RNAi that reversed chiral self-organisation upon for example latA or profilin1 RNAi treatment, depends on mDia1. In addition, a bit more discussion on this matter would also be highly appreciated.

- The authors previously published that the chiral self-organization of the actin cytoskeleton arises gradually over time. Therefore, quantifying the time-evolution of chirality in control and upon knockdowns could be an informative experiment to do.

- A related comment: For the analysis in this manuscript, the authors mostly used fixed cells. Can the authors exclude that the perturbations in which chirality is reduced, were in fact cells in which the onset of the chiral phase was postponed, but the chiral phase itself unaffected?

- Also related: Please specify more clearly in which experiments the authors used fixed cells, and in which experiments they relied on live cell imaging.

- Figure S6A: the figure legend says: 'Rescue of Profilin 1 knockdown cells by co-transfection with Pfn1-P2A-eGFP full-length plasmid is shown in lane 3 of western blot (upper blot).' However, the blot shows only 2 lanes; there seems to be one condition missing.

Minor comments:

The Grill group has previously looked at the effect of a number of actin regulators on actomyosin cortex structure and dynamics (Naganathan et al., Morphogenetic degeneracies in the actomyosin cortex, eLife 2018), including chiral movements. A number of the genes investigated here were also tested there, so could the authors comment on similarities/differences in the conclusions drawn?

In Figure 1, the authors could clarify whether the example without chirality shown in fig 1A is an outlier or not. As there is essentially no chirality in A this seems to be an outlier and not a representative image. Alternatively it could be that A and E are the extremes and that the data lies within them, please clarify..

For clarity of the figure, it would be nice to have the same x axes in figures 1d and 1h and to have them aligned such that comparison is easier.

There seem to be different dependencies on different formins for the single cell and collective cell chirality. Can the authors comment on this?

The actin cytoskeleton seems to be quite differently organized in the fluorescence micrograph of the profilin1 rescue experiment (figure 2D), when compared to the control. Can the authors comment on why that might be the case? Could this condition be a mixture of chiral and chordal, or chiral and linear (as classified in their 2015 paper). Might this be related to overexpression? It would be nice to see the western blot for profilin1 rescue.

Reviewer #3 (Remarks to the Author):

This is a very interesting manuscript that addresses the difficult problem of the origin and the relationship between single cell chirality and population-scale chirality. The article doesn't entirely solve the question but it represents a very significant advance in the field. It is I believe the first time that the role of actin is clearly identified in the supracellular chiral arrangement of cells. The screen of actin-associated proteins is very thorough and the implication of the actin cytoskeleton chirality at single cell level and tissue level is a major conclusion. The strong correlation between both scales (Fig. 4) is a definite landmark in this field. So, there is no doubt for me that this work deserves publication in Nature Communications. The following remarks are meant to clarify a few points, they should not prevent or unnecessarily delay this publication.

The authors address chirality but they implicitly do it with elongated cells that describe a nematic system. Two nematic systems in fact, the actin cytoskeleton at the single cell level and the cells themselves at the collective level. Yet, the authors do not discuss this important characteristics of these architectures. In particular, the individual cells show a +1 spiral centered defect and the confined cell population two -1/2 defects. Why choosing two different geometries (circle and rectangle) rather than comparing single cells on disks to cell populations on larger disks? The topological charge would then be the same (+1) in both cases.

This last remarks goes beyond the nematic aspect. The organization of the cell populations is very constrained by the nature of the rectangular confinement. By nature, the diagonal angle is fixed at +/- 26°. Why would the average angle in this population be a measurement of chirality? On the same line, I commend the authors for making quantitative measurements of chirality-related quantities (angles); however, although useful to compare different cells placed in the same conditions, these measurements are only relative. Would it be possible to define a system-free "absolute chirality"? (ie a quantity that underlies the descriptors used in the present study but doesn't depend on the system or its geometry).

Other works have used different criteria to characterize the nematic character of their system and have in particular measured the dynamic rotation of single cells or cell ensembles. Would such an approach be possible here?

How do the authors interpret the tilt of the stress fibers of cells plated on an elliptical domain? Can the interpretation in terms of active gels mentioned in the discussion (p18, l8-12) be used here as well?

The authors very openly admit that they don't understand everything in their experiments: Why doesn't the down regulation of the myosins affect chirality? What is the mechanism of chirality inversion resulting from some of the knockdowns or drugs? How does single cell chirality results in tissue-scale chirality? These are all important questions and I have no

doubt that the present paper will be the basis of further experimental and theoretical works that will help answering them.

Point by point answers to the referees:

Reviewer #1 (Remarks to the Author):

Cell and tissue-scale chirality, which results somehow from the scaling-up of molecular- and polymer-scale chirality, is fascinating and fundamental to living things. Tee, Bershadsky and colleagues had previously demonstrated a key role for the crosslinker alpha-actinin in single-cell chirality and provided a conceptual model of the accumulation of torsional stresses within actin networks generated by membrane-anchored formins and stored via actin crosslinking. Here, they further demonstrate molecular mechanism for the existence and chirality of the tilt of radial actin bundles in single cells cultured on circular patches. The movies of this process are very helpful and visually stunning.

The work is highly quantitative, careful and thorough. For example, the demonstration of each protein depletion's effect on other proteins of interest in addition to the target is much appreciated. That said, limited additional quantitation of their existing data with newly developed computer vision would add key insights into mechanism. In addition, the current version of the manuscript suffers from inadequate discussion of physical mechanism. I make some suggestions here for how the authors could fill out their discussion – I look forward to learning what they think.

Major points:

1) Radial actin bundles appear anchored to the nucleus, which itself swirls around the cell. What structure provides the pulling or pushing force for these various movements? Is generation of F-actin, spooling out from peripheral formins, sufficient? Or is the nucleus drawn through the cytoplasm by something else, taking the radial bundles with it? I doubt this latter idea, since in one of the two examples shown, the nucleus has a much more limited travel than the radial bundle apexes.

Response #1.1: The dynamics of the self-organisation of the actin cytoskeleton from radial symmetry to anti-clockwise swirling was reported earlier by us (see Figure 8c in Tee *et al.*, 2015). Briefly, unidirectional tilting of the radial fibres growing from focal adhesions converts the radial-centripetal movement of transverse fibres into spiral movement, resulting in anti-clockwise swirling actin flow. In our experiment, we found that enucleated cells retained the ability to organise chiral actin cytoskeleton in both control and latrunculin A-treated conditions (supplementary Fig. S8 and Movie S6). This suggests that movement of the nucleus is not the causal factor of radial fibre tilting, but more likely actin cytoskeleton swirling is driving nuclear movement. We have now emphasized this in the text of the Results (see page 11, lines 10-14). The detailed mechanism(s) of nuclear rotation, as well as the chiral movement of other cytoskeletal networks such as the cytokeratins (Jalal *et al.*, 2019) and microtubules (Tee *et al.*; Jalal *et al.*), and organelles such as the Golgi apparatus, the endoplasmic reticulum and the centrosomes (our unpublished observations), deserves special studies which are beyond the scope of the present work.

2) I don't agree that mDia1 depletion "abolished the chirality of microculture orientation" since there are still many micropatches with normal chirality, and there are many with strong but reversed chirality. The proportion of micropatches with no chirality is unchanged. When the distribution becomes multimodal, the mean angle is not particularly meaningful – the center of each peak should instead be reported.

Response #1.2: We agree with the reviewer that the distribution of the cell alignment angle in microcultures of mDia1-knockdown cells is bimodal. Following the reviewer's suggestion, we applied the "sum of 2 Gaussians distribution" fitting, implemented in GraphPad Prism 9 software, to the histograms representing mean nematic director angle and mean nuclei orientation of both the control siRNA- and mDia1 siRNA-treated microcultures (Fig. 1P). The results reveal that the distributions of mDia1 siRNA-treated microcultures, but not the control microcultures, can be represented as a sum of 2 Gaussians distribution with two different means (\pm s.d.), which are now indicated and shown in the figures in this revised manuscript (Fig. 1P and supplementary Fig. S4).

3) How do groups of cells acquire similar tilt? Are they pushing on each other or pulling? The authors' computer vision should be applied to their wonderful timelapse image series of these patches' evolution. Do the pointed-into corners (upper-right and lower-left in controls) lead the way, or do the upper-left and lower-right corners first get cleared out instead? Does the theory in Reference 9 make any predictions that the authors can test?

Response #1.3: The questions raised by the reviewers are indeed very interesting and deserved both experimental clarification and theoretical explanation. In particular, the nature of contact interaction between cells and the role of cadherin-mediated adhesion in the course of alignment should be determined. We hope that our on-going studies will help to resolve these questions. We however, think that detailed description of reorganisation and alignment of cells on rectangular substrates will comprise materials for a separate publication. The theory in Reference 9 (Duclos *et al.*, 2018) nicely explain mutual orientation of chiral nematic objects, while not specifying the nature of their interaction with the micropatterned substrate and each other. Since our present study reveals a strong correlation between the chiral tilting of the actin fibres inside the cells and chiral mutual orientation of the cells in cell group confined to rectangular micropattern, the goal of our on-going study is to elucidate the mechanism connecting these two processes. The essence of our model (which we have now briefly mentioned in the Discussion (see page 20, lines 6-20) is that the origin of the multicellular chiral pattern stems from interactions of the individual cells with the boundaries of the adhesive pattern. A cell migrating from the interior of the pattern toward the boundary tends to orient in such a way that the cell polarises and migrates anti-clockwise around the pattern's boundary due to the likely connection between the anti-clockwise tilting radial fibres and this polarisation and migratory behaviour at the boundary.

4) The authors should connect their work here to the examination of the separate contributions of torque generation (formin-based polymerization, myosin-based translocation) versus torque storage (crosslinkers, membrane anchoring)? Their candidate list contains both.

5) Related: The similar effect of depleting formins and the Arp2/3 complex is puzzling. The authors say that the proteins whose depletion phenocopies formin depletion "function in cooperation or competition with formins" - why would these two opposing activities have the same effect? Some discussion should be provided. None of the treatments should change the chirality of F-actin, so how is cellular chirality reversed? Could Arp2/3 depletion and other treatments that attenuate actin polymerization affect cell chirality by changing the degree to which the formin stays associated with the barbed end, thus changing its ability to store torque into the filament. If the formin slipped/swiveled as it added actin subunits, torque would not be stored in the filament. Perhaps there is a formin-slippage regime

wherein it essentially slips the opposite direction, building torque of the opposite handedness into the filaments. I'm having a hard time describing this, but it would be similar to the phenomenon of how a car's hubcap can appear to turn backwards with respect to the wheel's revolution, based on the framerate of a camera or your eye being just shorter than the revolution rate. Perhaps the direction and magnitude of torque on F-actin could be tested by assessing relative susceptibility to cofilin severing, or perhaps F-actin probes (CH domains, LifeAct, etc) have different affinities to super-coiled vs under-pitched F-actin (I am taking the liberty to invent terms for actin that has more or less torque.) The authors should include substantial discussion of how their perturbations can cause cellular chirality reversal; I'm keen to know what they think of these suggestions.

Responses #1.4 and #1.5: We are grateful to this reviewer for his/her interest to our ideas concerning a unifying explanation of our experimental results. If we had such a unifying model, we most definitely would have shared it with the readers. Unfortunately, our present knowledge of the functions of the actin regulators involved in chirality determination seems to be insufficient to provide such a unifying explanation. However, we can comment on the particular questions raised by this reviewer. Part of these comments are now included in the Discussion (see page 18, line 14 to page 19, line 16).

i. One of the general conclusions from this study is that cellular and multicellular chirality is not a consequence of the activity of a single actin-associated protein or even one class of proteins (formins, Myo-I isoforms). Instead, the chiral behaviour emerges from coordinated activities of actin nucleators, regulators of actin polymerisation/depolymerisation and actin crosslinkers such as alpha-actinin. We think that the key to understanding the molecular mechanisms of converting the microscopic chirality of individual actin filaments into the cell-level chirality of the cytoskeleton is in microscopic architecture of the actin-based structures.

ii. The possible role of Arp2/3 complex. Reduction of the Arp2/3 function may promote actin polymerising functions of formins since more monomeric actin will be available for them (Suraneni *et al.*, 2012). On the other hand, Arp2/3 together with SPIN90 protein can promote mDia1, but no mDia2, -driven actin polymerization (Cao *et al.*, 2020). Altogether, these effects could create imbalance in the participation of different formins in the assembly of cellular actin structures, reducing the impact of mDia1, the main chirality factor.

iii. We agree with this reviewer that the reversal of chirality direction upon genetic manipulations or pharmacological treatments is a key phenomenon still lacking an adequate explanation. In our previous paper, we speculated that the torque induced by formin-driven rotation of trapped filament can occasionally be released (when elastic energy of the system approaches some threshold), leading to the rotation of filament in the opposite direction (Shemesh *et al.*, 2005). This resembles the idea mentioned by this reviewer regarding "formin-slippage regime". Indeed, in our paper (Tee *et al.*, 2015) we have mentioned that such consideration could explain the reversal of cell chirality in alpha-actinin1 overexpressing cells. In recently published papers (Li and Chen., 2021 and 2022), this model was elaborated and applied to the explanation of the phenomenon of chirality reversion. Verification of this model will require substantial further work. We have now included these references to our reference list, refs 38 to 40.

iv. Another model connects the direction of cell chirality with the structural organisation of the interactions between radial and transverse actin fibres. We posited previously that the initial breaking of the left-right symmetry starts when the radial actin bundle rotating unidirectionally around its long axis begins to 'roll' on circumferential transverse actomyosin

structures in a 'rack-and-pinion' mechanism (Tee *et al.*, 2015; Mogilner and Fogelson, 2015). If the circumferential structures are 'above' the radial actin bundles, then the clockwise-rotation of the bundles (if one looks along the bundle axis from the barbed ends at focal adhesions) results in anti-clockwise swirling in the cell. However, if the circumferential structures are 'below' the radial actin bundles (between the substrate and bundles), then the clockwise-rotation of the bundles should produce clockwise swirling. Consequently, the lateral cooperative interactions between the bundles organise the cell-scale chirality pattern. It is likely that some perturbations of the actin dynamics and/or crosslinking could change the mutual position of radial and transverse fibres, thereby changing the direction of cell chirality. The weakness of this explanation, of course, is that precise predictions are hard to make. Future structural study will evaluate this hypothesis.

Minor points:

6) I'm not sure it's accurate to say that "the majority of multicellular organisms demonstrate approximate bilateral symmetry" since that is not true for any land plants or fungi that I know of, or many animals including echinoderms, some gastropods, and probably many more of which I am less aware. You could say "many animals demonstrate approximate bilateral symmetry."

Response #1.6: We thank the reviewer for highlighting this. We have changed our text accordingly (see page 3, line 2).

7) Schonegg, Hyman and Wood needs to be referenced with 11-14, as they were the first to show what is attributed to reference 12 here.

Response #1.7: We are grateful to this reviewer for directing us to the proper reference and have now included it in revised manuscript as ref #13.

Reviewer #2 (Remarks to the Author):

In this manuscript Tee et al use an assay previously developed by this group in which fibroblasts are seeded on micro patterned circular surfaces to systematically study the role of actin regulators in the emergence and handedness of cellular chirality. In addition, they have extended their findings to the chirality of cell collectives on rectangular surfaces. They found further evidence that supports the hypothesis that chirality of cell collectives arises as a result of cell-intrinsic chirality. Overall the work is exciting, provides novel insights into the molecular control of cell chirality, and makes the perhaps central claim that cellular chirality emerges in nontrivial ways from molecular constituents, and as such can be tuned in several ways. The conclusions are solid and backed up by experimental results (with some exceptions, see below). This paper should be published once the main concerns below are addressed.

Main concerns:

1) Did the authors rule out that the image analysis pipeline used to segment radial actin fibers introduces bias? For example, if an RNAi decreases bundle width or curvature, is it segmented better/worse? Does the method miss bundles if their intensity drops in a given RNAi condition? Or vice versa, does the method pick up more bundles if the background level of F-actin is reduced?

Response #2.1: We thank the reviewer for the important comment concerning our methodology. We agree with the reviewer that the method of assessment of the radial fibres tilt angles should be robust in the sense that it should not depend on non-relevant changes in the actin cytoskeleton. In particular, in our procedure of the AI-algorithm training, we trained the algorithm to weight equally both thick and thin fibres (see new supplementary Fig. S1, A and B and Text page 6, lines 13-15). As shown in Fig. S1B, the algorithm positively identified RFs of varied fluorescence intensity and efficiently ignored the fluorescence signals from their neighbouring transverse actin fibres. In the determination of the tilting angle of the radial fibres, our measurement of the radial fibres angles was performed on the short linear segments of fibres located inside the narrow concentric rings (see supplementary Fig. S1C), so that the radial fibre curvature did not affect the results. Thus, we believe that our methodology is robust to the concerns mentioned by the reviewer. Overall, our combination of the automated AI-based algorithm and careful visual inspection of the morphologies of the actin cytoskeleton allows us to properly assess the parameter characterising the radial fibre tilting.

2) Some RNAi's, for example profilin1, cofilin1/2 and CapZ, result in only small reduction in protein levels, as judged by eye from the western blots, i.e. the ratio between loading control and protein of interest. On top of this, the RNAi effect on cell chirality of profilin1 KO alone and CapZ KO alone seems, by eye, weaker in figure 3 (J and K respectively) than in figure 2 (B and E respectively). Actually, by eye the distribution of angles seems around zero, e.g. no chirality, in figure 3. The authors should verify with statistical tests whether these distributions are different from zero. Similarly, authors should address whether the distributions of profilin1 RNAi in fig 2 and 3, and CapZ RNAi in figure 2 and 3 are statistically the same or different.

Response #2.2: We thank the reviewer for highlighting the gaps in statistical analysis in the first version of this manuscript. This prompted us to systematically apply statistical tests to

all the data represented in histograms. We have performed two types of statistical tests to facilitate our data interpretation – (1) Kruskal-Wallis test for multiple comparisons across groups to report any significant difference between groups, and (2) Wilcoxon signed-rank test to estimate the difference of median values of samples from zero. These were performed not only to the histograms as highlighted by the reviewer in this question, but were done for all the relevant histograms that we presented in the manuscript and the results are now reported in new supplementary Table S1.

To respond specifically to the histograms referred to by the reviewer in this comment, the distribution of angles in figures 2B, 2E, 4J (previously 3J) and 4K (previously 3K) are significantly different from zero (using Wilcoxon signed-rank test; see Table S1, lines 20, 24, 94 and 98). In addition, the distributions of angles in profilin1 siRNA-treated cells represented in figures 2B, 3B (new figure) and 4J (previously 3J) are not statistically significantly different as determined by Kruskal-Wallis test (see Table S1 lines 230-232). Similarly, the distributions of angles in CapZ β siRNA-treated cells in figures 2E and 4K (previously 3K) are also not statistically significantly different (see Table S1 lines 236).

Concerning the first part of this question, we have now indicated the fold changes in the protein levels for each knockdown as determined by western blots (see Fig. 3H; S2, D and F; S5, F and G; S6E; S7, B to D; S10E and S11, B and C). We confirmed that significant effects on cellular chirality and cell group alignment were observed despite the seemingly moderate reduction ~40-70% in the level of profilin 1, cofilins 1&2 and CapZ β .

3) Related to previous point: If the distributions of profilin1 and capZ in figure 2 and 3 are different, this would suggest that the knockdown efficiency is different in different experiments. Can the authors comment on this? If true, this would raise a couple of additional questions: 1) Cofilin1/2 knockdown decreases, but does not reverse chirality. This might be due to poor knockdown efficiency at the protein level. 2) The double knockdowns in fig 3 (for example profilin1 with alpha actinin knockdown) might have different knockdown efficiency than either single knockdown, which could also explain the results in case of profilin1+alphaAct1 (e.g. no difference from control because of low knockdown efficiency).

Response #2.3: As we mentioned in our **Response #2.2**, we have now included quantitative data showing the level of expression of the protein of interest upon their knockdown in each of our experiment. Indeed, the efficiencies of knockdown are varied among different experiments. However, despite the different knockdown efficiencies, in all the experiments, we found that profilin-1 and CapZ β knockdown cells demonstrated reversed cell chirality (see **graphs** Fig. 2F and 3G and **histograms** Fig. 2B, 2E, 3B, 4J and 4K). These distributions represented by the histograms were not found to be significantly different between experiments (as mentioned in **Response #2.2**, see Table S1 lines 230-232 and 236).

We had classified the effects of different knockdowns as **reducing** or **reversing** chirality based on the shape of their graphs. mDia1 knockdown efficiency is >90% in different experiments (see Fig. 3H, S2D and S11B) and mDia1-knockdown cells demonstrated **reduced** but not reversed chirality as represented by the graphs (see Fig. 1L, 3G and S2B). Specifically, these graphs show that the mean values of RF tilt angles were smaller than in control but still positive at varied distances from the cell edge (see Fig. 1L, 3G and S2B). On the other hand, similar graphs for knockdowns that we classified as **reversing** chirality direction showed negative values at varied distances from the cell edge (see Fig. 2F, 2H,

3G, 4B and S8E). Based on this definition, cofilins1&2 or ARPC2 knockdown cells were classified as **reducing** chirality, while profilin-1 or CapZ β knockdowns and latrunculin A treatment as **reversing** the chirality.

The reviewer further suggested that the reduction in chirality observed in cofilins1&2 knockdown cells could result from either a reduction in the emergence of anti-clockwise chirality or an induction of clockwise chirality. In the latter scenario, the histogram depicting RF tilt angle distribution should trend towards a bimodal distribution – with 2 means, 1 at positive (depicting anti-clockwise chirality) and another at the negative (depicting clockwise chirality) value. However, the corresponding histogram (see Fig S5C) hardly represent bimodal distribution since, in particular, it cannot be fitted as a sum of 2 Gaussian distributions.

Finally, we have introduced new western blot data that reports on the efficiency of profilin1 and alpha-actinin1 individual knockdowns and their double knockdowns for three repeats (see Fig S7C). In summary, to answer the reviewer's concerns, the efficiency of gene knockdowns in double knockdown experiments was not diminished as compared to individual knockdown experiments. Therefore, in Fig. 4, the absence of chirality reversal observed in Profilin1 and alpha-actinin1 double knockdowns could not be due to the low knockdown efficiency of Profilin1 under this condition but rather points towards a critical role for alpha-actinin1 in chirality reversal induced upon Profilin1 knockdown.

4) In case of cofilin1/2 KO there appears to be a 1.5 fold up regulation of alpha-actinin (fig S10). Does this affect the conclusions drawn on the effect of cofilin1/2?

Response #2.4: Indeed, level of alpha-actinin expression can in principle affect chirality as we reported previously (Tee *et al.*, 2015) and in this paper (see Fig. 4B and S9A). Therefore, we checked whether the increase in mRNA transcripts of alpha-actinin1 (see Fig. S11A) also resulted in an increase in alpha-actinin1 protein level (see Fig. S11, B and C). Our new western blot results show that alpha-actinin1 protein level was not elevated in cofilins 1&2 knockdown cells (see Fig. S11C). Similarly, we also checked the alpha-actinin1 protein level in mDia1 knockdown cells since the RNAseq data also suggested a slight increase in alpha-actinin1 transcript levels following mDia1 knockdown. Our new western blot results show that alpha-actinin1 protein level was also not elevated in mDia1 knockdown cells (see Fig. S11B).

5) The finding that profilin1, but not Formin knockdown, results in chirality reversals is very interesting. According to the author's previous explanation for chirality reversals (Tee *et al* 2015), both clockwise and counter-clockwise chirality depends on Formins. In one case the authors hypothesised this is due to the 'stair-stepping' mode of Formin while in the other it is due to 'screw-stepping' mode of Formin. It would be very informative to the field if the authors could verify by RNAi that reversed chiral self-organisation upon for example latA or profilin1 RNAi treatment, depends on mDia1. In addition, a bit more discussion on this matter would also be highly appreciated.

Response #2.5: We are very grateful to this reviewer for attracting our attention to this matter and we indeed obtained informative results following the reviewer's suggestion. The results have now been composed into a new figure and introduced as a part of the main figures – new Figure 3 and mentioned in the Results (see page 12, lines 3-12). We followed

the reviewer's suggestion to check whether formin mDia1, which plays a key role in the development of anticlockwise chirality could also participated in the emergence of clockwise chirality in profilin1-knockdown cells and latrunculin-treated cells. Our results revealed that mDia1 and profilin-1 double knockdown cells demonstrated clockwise chirality as efficiently as profilin-1 only knockdown cells (histograms from both conditions, Fig. 3B vs. 3E, are not significantly different, see Table S1, line 48). Our western blot data showed that the knockdown efficiencies of mDia1 and profilin1 in double knockdowns cells are comparable to their individual knockdown (see Fig. 3H), eliminating the possibility that the lack of difference seen was due to inefficient knockdowns. Similarly, treating mDia1 knockdown cells with latrunculin A also induced reversed chirality in these cells (histograms from both conditions, Fig. 3C vs. 3F, are not significantly different, see Table S1, line 49). Overall, these new results suggest that mDia1 formin is indispensable for the anti-clockwise chirality, but not for clockwise chirality induced by reduction of profilin 1 protein level and latrunculin A treatment. As this reviewer mentioned, a model we proposed in our earlier publication suggested that mDia1 (or other formins) can be responsible for both anti-clockwise and clockwise chirality. Here, we unequivocally showed that at least for mDia1 it is not the case.

6) The authors previously published that the chiral self-organization of the actin cytoskeleton arises gradually over time. Therefore, quantifying the time-evolution of chirality in control and upon knockdowns could be an informative experiment to do.

7) A related comment: For the analysis in this manuscript, the authors mostly used fixed cells. Can the authors exclude that the perturbations in which chirality is reduced, were in fact cells in which the onset of the chiral phase was postponed, but the chiral phase itself unaffected?

8) Also related: Please specify more clearly in which experiments the authors used fixed cells, and in which experiments they relied on live cell imaging.

Responses #2.6: Our paper contains the graphs showing the changes with time of the tilt of stress fibres in elliptical cells (see Fig. 6, G to K). To provide the live imaging data for all types of conditions (>35) we used in this study is however technically difficult and beyond the scope of present work. Moreover, we would like to emphasize that using LifeAct required for visualisation of actin in live imaging experiment can produce some artefacts as it was previously published (Courtemanche *et al.*, 2016). Indeed, we have noticed that under some circumstances the LifeAct-labelled cells do not demonstrate the effects that we observed studying non-labelled cells in which actin was visualised by phalloidin after fixation. We have highlighted these discrepancies in the text (see page 10, lines 15-19). Actually, our system which appears to be very sensitive to many types of actin perturbations could be a good model to characterise possible side effects of different reagents used for live actin cytoskeleton imaging. These studies however, are beyond the scope of present work.

Response #2.7: The question reviewer asked does not actually require the live imaging and can be addressed by fixing cells at later time points than we are usually using (6 hrs post cell seeding for individual cells and 48 hrs post seeding for cell collectives.). We selected the 6 hr time point for the assessment of single cell chirality because as we have shown in Tee *et al.* (2015), at later timepoints, radial cells can develop the system of long parallel stress fibres. Thus, we agree with this reviewer that, strictly speaking, we cannot distinguish between factors that abolish cell chirality and factors that delayed the onset of chirality phase. It can happen, for example in the situation of ARPC2 and cofilins1&2 knockdowns,

since such cells still demonstrated the chirality in cell collectives which was assessed at 48hrs after cell plating. We have now included this possibility in the text (see page 9, lines 9-11). Systematic investigation of the long-term effects of all the knockdowns used on actin cytoskeleton evolution is technically difficult and beyond the scope of present study.

Response #2.8: In the revised version, we more clearly indicated which cells – either stained with phalloidin after fixation (fixed cells) or transfected with LifeAct (live cell imaging) – were used for each type of measurements in the Figure Legends. Specifically, in the figure legends for fixed cells, we have indicated that histograms or graphs are obtained from “cells fixed at 6 hours after plating”. For data using live cell imaging, we have indicated that histograms or graphs are obtained from “cells labelled with LifeAct and imaged for 12-16 hours”.

9) Figure S6A: the figure legend says: ‘Rescue of Profilin 1 knockdown cells by co-transfection with Pfn1-P2A-eGFP full-length plasmid is shown in lane 3 of western blot (upper blot).’ However, the blot shows only 2 lanes; there seems to be one condition missing.

Response #2.9: We thank the reviewer for highlighting this mistake. The correct blot that matches the figure legend is now depicted (see Fig. S7B).

Minor comments:

10) The Grill group has previously looked at the effect of a number of actin regulators on actomyosin cortex structure and dynamics (Naganathan et al., Morphogenetic degeneracies in the actomyosin cortex, eLife 2018), including chiral movements. A number of the genes investigated here were also tested there, so could the authors comment on similarities/differences in the conclusions drawn?

Response #2.10: We thank the reviewer for drawing our attention to the above paper. Naganathan *et al* (2018), analysing cortical flow on a large scale and its response to multiple perturbations, coined the notion of a ‘morphogenetic degeneracy’, where distinct activities at the molecular scale contribute to similar physical activities at larger scales (one of such activities was, in fact, the flow chirality). Our results agree with this notion: we observed that diverse molecular perturbations of actin dynamics change the prominence of the chiral cell-scale organisation or change the sign of chirality. Some of the actin regulators identified in our work (profilin, Arp2/3, cofilin, capping protein) were also mentioned in Naganathan (2018). We added the note to this effect in the Discussion (see page 17, line 21-23). Furthermore, it is worth noting that in their latest publication (Middelkoop *et al.*, 2021) reporting on the same aforementioned phenomenon of cortical flow in *C.elegans* zygote also strongly points towards a formin-driven mechanism underlying chiral actomyosin-based cortical flow in *C. elegans*. We have extensively referred to this important study in our paper (Middelkoop *et al.*, 2021, ref #23 in Reference list).

11) In Figure 1, the authors could clarify whether the example without chirality shown in fig 1A is an outlier or not. As there is essentially no chirality in A this seems to be an outlier and not a representative image. Alternatively it could be that A and E are the extremes and that the data lies within them, please clarify..

Response #2.11: We thank the reviewer for bringing this to our attention. Indeed the population of control cells at 6 hours after plating contained some fraction of “non-chiral” cells with non-tilted radial fibres. The cells shown in Fig. 1A belongs to this category. In majority of the cells however, the radial fibres are tilted in anti-clockwise direction as shown in Fig. 1E. In our Figure 1, we selected the cells of these two types to illustrate our methodology of quantitative assessment of cell chirality based on measurement of radial fibre tilt. In the revised version, we have changed the legend of figure 1 and explained this clearer.

12) For clarity of the figure, it would be nice to have the same x axes in figures 1d and 1h and to have them aligned such that comparison is easier.

Response #2.12: We thank the reviewer for the suggestion. Figure 1D and 1H now has the same x-axes (see revised Fig 1D and 1H).

13) There seem to be different dependencies on different formins for the single cell and collective cell chirality. Can the authors comment on this?

Response #2.13: We have now provided a new correlation graph depicting the rank correlation between actin cytoskeleton chirality in individual cells and collective cell chirality in microcultures for the group of formin family members proteins that we have examined in this study (see Fig. S12B). For majority of the formin knockdowns studied, the rank of their effect on tilt of radial fibres in individual cells correspond to the rank of effect on the tilt of cell alignment. The dependency between actin cytoskeleton chirality and collective cell chirality is different for knockdowns of formins, FMN2 and INF2 as compared to the bulk of formins. Unlike all other formins, INF2 formin knockdown increased the collective cell chirality without affecting individual cell chirality, while FMN2 somewhat decreases individual cell chirality but increases the collective chirality. However for other eight formins, the relationship between effects on single cell chirality and collective cell chirality is highly correlative (Spearman $r = 0.9500$, 9 XY-pairs).

14) The actin cytoskeleton seems to be quite differently organized in the fluorescence micrograph of the profilin1 rescue experiment (figure 2D), when compared to the control. Can the authors comment on why that might be the case? Could this condition be a mixture of chiral and chordal, or chiral and linear (as classified in their 2015 paper). Might this be related to overexpression? It would be nice to see the western blot for profilin1 rescue.

Response #2.14: We have selected a new image to better represent the rescued cells in Fig. 2C (previously 2D). In the previously selected fluorescence image of profilin1 knockdown-and-rescued cell, the reviewer noted the presence of ventral stress fibres seen in the maximum z-stack projection of the actin cytoskeleton in that particular example. Ventral stress fibres are not considered in our classification of chiral, chordal or linear organisation of actin organization pattern as reported in Tee *et al.*, 2015 as they are found randomly organised on the most ventral surface of the cells and do not rise dorsally typical for radial fibres. Furthermore, ventral stress fibres are not unique structures found in profilin1 knockdown cells and can be found in control cells and other conditions too. Therefore, we opted to change the image (Fig. 2C). Concerning the western blot for profilin1 rescue, we have now included it in Fig S7B.

Reviewer #3 (Remarks to the Author):

This is a very interesting manuscript that addresses the difficult problem of the origin and the relationship between single cell chirality and population-scale chirality. The article doesn't entirely solve the question but it represents a very significant advance in the field. It is I believe the first time that the role of actin is clearly identified in the supracellular chiral arrangement of cells. The screen of actin-associated proteins is very thorough and the implication of the actin cytoskeleton chirality at single cell level and tissue level is a major conclusion. The strong correlation between both scales (Fig. 4) is a definite landmark in this field. So, there is no doubt for me that this work deserves publication in Nature Communications. The following remarks are meant to clarify a few points, they should not prevent or unnecessarily delay this publication.

Response: We are grateful to this reviewer for the favourable evaluation of our work.

1) The authors address chirality but they implicitly do it with elongated cells that describe a nematic system. Two nematic systems in fact, the actin cytoskeleton at the single cell level and the cells themselves at the collective level. Yet, the authors do not discuss this important characteristics of these architectures. In particular, the individual cells show a +1 spiral centered defect and the confined cell population two -1/2 defects. Why choosing two different geometries (circle and rectangle) rather than comparing single cells on disks to cell populations on larger disks? The topological charge would then be the same (+1) in both cases.

Response #3.1: We note that the topological defects for multiple cells on rectangular patterns are likely accidental; they are the consequences of the particular biased cell movements described in response to Reviewer 1 (see **Response #1.3**). Similarly, for individual scale the topological defect at the centre is simply due to the nucleus presence there. We believe that in our systems the topological defects do not capture the essence of the molecular mechanisms of chirality. On the single cell scale, the analogy between actin cytoskeleton and a nematic system has to be used with caution – there are very explicit and dynamic molecular microscopic patterns in the actin network that are not captured by the unifying notion of a nematic gel. We have not put the single cells on the rectangular adhesive patch for a single reason – strong and large focal adhesions would be expected to form in the corners of such cell, which was reported often, making certain symmetric stress fibre pattern which will obscure chirality. However, we presented the data on individual cells at elliptical micropattern which, in some sense, could be analogous to the rectangular micropattern for the cell groups. Putting multiple cells onto a circular pattern could reveal chirality, if its appearance was macroscopic rotation of the cellular groups, which is not always the case (Guillamat *et al.*, 2022). The pattern that was used by several independent studies to demonstrate the chirality in cell groups was the stripes of indefinite length (Chen *et al.*, 2012; Duclos *et al.*, 2018). Behaviour of cells on our rectangular patterns resembled that of cells on stripes and revealed the chirality in cell groups sufficiently well. Ultimately, our goal in this study, as far as the multicellular system is concerned, was simply to establish if the chirality evolves at all, and if the sign of the chirality for multicellular system correlates with that for single cell, for which the rectangular patterns served their purpose. We do, however, plan experiments with multiple cells on an annulus pattern and on a rectangle with smooth corners, thanks to incisive comments of the reviewer.

2) This last remarks goes beyond the nematic aspect. The organization of the cell populations is very constrained by the nature of the rectangular confinement. By nature, the diagonal angle is fixed at $\pm 26^\circ$. Why would the average angle in this population be a measurement of chirality? On the same line, I command the authors for making quantitative measurements of chirality-related quantities (angles); however, although useful to compare different cells placed in the same conditions, these measurements are only relative. Would it be possible to define a system-free "absolute chirality"? (ie a quantity that underlies the descriptors used in the present study but doesn't depend on the system or its geometry).

Response #3.2: The 'absolute chirality' is rather simple to define: it is the left-right symmetry break. In this sense, the preference for "И" pattern compared to "N" pattern is the absolute chirality. The 26° angle is accidental to the experimental design and its exact value is not important. However, note that there is no chirality on square patterns (45° angle), so we posit that the ratio of the long to short length of the rectangle side is an important factor. We cannot make the angle of the diagonal too close to 45° , otherwise the effect disappears. We cannot make it too close to 0° either – then the system becomes effectively 1d. Something between 15° and 35° is optimal to illustrate the effect. Going to the single cell level, again, simply left-right symmetry break is the qualitative definition of the chirality. However, reporting the tilt angle is still a useful measure of the 'magnitude' of chirality, or simply comparing quantitatively cell-level effect of a molecular state of the cell.

3) Other works have used different criteria to characterize the nematic character of their system and have in particular measured the dynamic rotation of single cells or cell ensembles. Would such an approach be possible here?

Response #3.3: As we said above, there are certainly useful analogies with the nematic systems: when we deal with elongated cells, nematic order parameters can be used to organise the data; the application of the nematic language to the description and analysis of the dynamics of the actin cytoskeleton inside cell confined to circular island is less obvious. In the current study, we feel that we would not gain much understanding if we use this technique. We do plan, however, to explore the nematic-style models in the future. Ultimately, our goal is to understand molecular mechanisms of single and collective cell chirality, and acknowledging that we are dealing with a system with nematic features is but the first step toward that goal.

4) How do the authors interpret the tilt of the stress fibers of cells plated on an elliptical domain? Can the interpretation in terms of active gels mentioned in the discussion (p18, l8-12) be used here as well?

Response #3.4: Observation of the dynamics of actin cytoskeleton development in cells on an elliptical substrate revealed that in this process, the chiral tilting of radial fibres (typical for circular cells) is also playing a dominant role. In elliptical cells, majority of the focal adhesions and radial fibres growing from them are formed at vertex regions of the ellipse. These radial fibres undergo unidirectional tilting similar to the radial fibres in circular cells. Sometimes, the radial fibres growing towards the centre of the cell from 2 opposite vertices fused with each other forming a long fibre tilted relatively to the long axis of the ellipses. We believe that such organisation of the stress fibres in elongated elliptical cells is an important factor for chiral mutual alignment of multiple cells on rectangular micropattern

even though the mechanism of translation of stress fibres tilting in individual cells into chiral nematic organisation of cells on rectangular substrate require further investigation.

The authors very openly admit that they don't understand everything in their experiments: Why doesn't the down regulation of the myosins affect chirality? What is the mechanism of chirality inversion resulting from some of the knockdowns or drugs? How does single cell chirality results in tissue-scale chirality? These are all important questions and I have no doubt that the present paper will be the basis of further experimental and theoretical works that will help answering them.

Response: We thank this reviewer again for the encouraging evaluation of our work. We agree with this reviewer in his/her selection of most interesting questions that deserve further experimental investigation and theoretical analysis.

References:

Cao, L. et al. SPIN90 associates with mDia1 and the Arp2/3 complex to regulate cortical actin organization. *Nat Cell Biol* 22, 803-814 (2020).

Chen, T.H. et al. Left-right symmetry breaking in tissue morphogenesis via cytoskeletal mechanics. *Circ Res* 110, 551-559 (2012).

Courtemanche, N., Pollard, T.D. & Chen, Q. Avoiding artefacts when counting polymerized actin in live cells with LifeAct fused to fluorescent proteins. *Nat Cell Biol* 20 18, 676-683 (2016).

Duclos, G. et al. Spontaneous shear flow in confined cellular nematics. *Nat Phys* 14, 728-732 (2018).

Guillamat, P., Blanch-Mercader, C., Pernollet, G., Kruse, K. & Roux, A. Integer topological defects organize stresses driving tissue morphogenesis. *Nat Mater* 21, 588-597 (2022).

Jalal, S. et al. Actin cytoskeleton self-organization in single epithelial cells and fibroblasts under isotropic confinement. *J Cell Sci* 132 (2019).

Li, X. & Chen, B. How torque on formins is relaxed strongly affects cellular swirling. *Biophys J* 121, 2952-2961 (2022).

Li, X. & Chen, B. Mobility of Alpha-Actinin Along Growing Actin Filaments Might Affect the Cellular Chirality. *Journal of Applied Mechanics* 88 (2021).

Middelkoop, T.C. et al. CYK-1/Formin activation in cortical RhoA signaling centers promotes organismal left-right symmetry breaking. *Proc Natl Acad Sci U S A* 118 (2021).

Mogilner, A. & Fogelson, B. Cytoskeletal chirality: swirling cells tell left from right. *Curr Biol* 25, R501-503 (2015).

Shemesh, T., Otomo, T., Rosen, M.K., Bershadsky, A.D. & Kozlov, M.M. A novel mechanism of actin filament processive capping by formin: solution of the rotation paradox. *J Cell Biol* 170, 889-893 (2005).

Suraneni, P. et al. The Arp2/3 complex is required for lamellipodia extension and directional fibroblast cell migration. *J Cell Biol* 197, 239-251 (2012).

Tee, Y.H. et al. Cellular chirality arising from the self-organization of the actin cytoskeleton. *Nat Cell Biol* 17, 445-457 (2015).

REVIEWERS' COMMENTS

Reviewer #1 (Remarks to the Author):

I appreciate the comprehensive, thoughtful, and impressive work the authors did to prepare the revision. It is a masterful work, highly deserving of publication in Nature Communications.

Reviewer #2 (Remarks to the Author):

The authors have done a laudable job at addressing all ours and all other referees concerns. This is a nice pice of work and the manuscript should be published as is.

Point by point answers to the referees:

REVIEWERS' COMMENTS

Reviewer #1 (Remarks to the Author):

I appreciate the comprehensive, thoughtful, and impressive work the authors did to prepare the revision. It is a masterful work, highly deserving of publication in Nature Communications.

Reviewer #2 (Remarks to the Author):

The authors have done a laudable job at addressing all ours and all other referees concerns. This is a nice pice of work and the manuscript should be published as is.

Response: We thank both reviewers for the very encouraging evaluation of our work.